# FLOWAGENT: A NEW PARADIGM FOR WORKFLOW AGENT

## ABSTRACT

Combining workflows with large language models (LLMs) allows LLMs to follow specific procedures, thereby extending their application to more real-world scenarios. However, incorporating workflows often compromises the flexibility of LLMs. For example in the case of Task-Oriented Dialogue (TOD), workflow atomize the function of LLM while programmatically imposing restrictions on execution path making the dialogue obstructed and less flexible when facing out-of-workflow (OOW) queries. Prompt-based methods offer soft control but sometimes fail to ensure procedure compliance. This paper introduces a new agent paradigm to address this challenge. Specifically, we first propose a novel Procedure Description Language (PDL) that integrates the flexibility of natural language and the precision of code for workflow expression. Additionally, we present a comprehensive framework that enables LLM to handle OOW queries while keeping execution safe with a series of controllers for behavioral regulation. This includes pre-decision and post-decision methods, where the dependency relationships between workflow nodes are modeled as a Directed Acyclic Graph (DAG) to validate node transitions. Beyond the primary objective of compliance considered in previous work, we introduce a new approach to evaluate the agent's flexibility in OOW situations. Experiments on three datasets demonstrate that FLOWAGENT not only adheres well to workflows but also responds better to OOW queries, showcasing its flexibility. Furthermore, exploration on WikiHow data confirms that the PDL effectively represents broader formats of workflow, inspiring further research on workflow-based QA tasks. [1]

## 1 INTRODUCTION

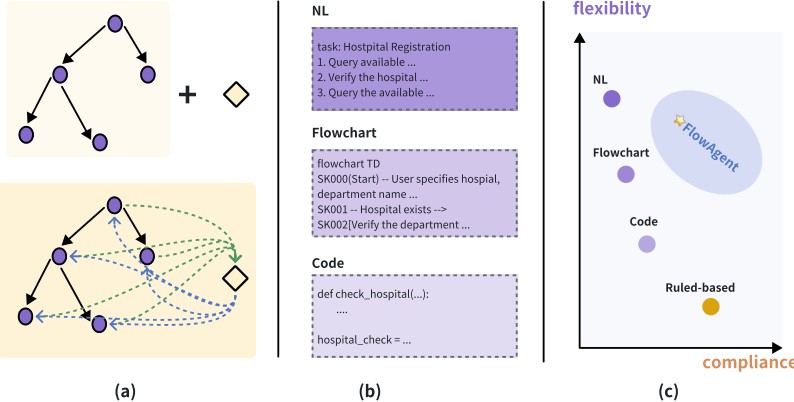

Figure 1: Comparison of different forms of workflow agents. (a) Rule-based workflows. (b) Representations of workflows in text forms. (c) Conceptional comparison of workflow methods in terms of flexibility and compliance.

With the enhanced understanding and reasoning capabilities of large language models (LLMs), pretrained LLMs are increasingly being utilized in dialogue systems (He et al., 2022; Bang et al., 2023).

---

[1] Code&prompts are available at https://anonymous.4open.science/r/FlowAgent-DE68/

Compared with traditional chatbots, LLMs can interact more flexibly with users to address diverse needs, leveraging the vast amount of commonsense knowledge stored in their parameters (Yi et al., 2024). However, in real-world applications, we often expect chatbots to follow specific rules and procedures to perform certain tasks (e.g., guiding users to make an appointment for appropriate hospitals, departments, and doctors(Mosig et al., 2020; He et al., 2022)). The procedures that must be followed through dialogues are known as *workflows*. LLMs, acting as *workflow agents*, assist users via conversations and invoke relevant tools to fulfill requests (Xiao et al., 2024).

Existing research can be broadly classified into two categories: rule-based and prompt-based methods. **Rule-based** methods (Coze, 2024; Dify, 2024; Flowise, 2024) control the conversation between the agent and the user through deterministic programs, modeling the progress of dialogue as state transitions within a graph composed of nodes representing different dialogue states, as shown in the upper part of Fig. 1(a). In this approach, the LLM functions as a node within the graph and cannot control the entire conversation flow. As a result, this method provides high compliance but often at the expense of the LLM's inherent *flexibility*. As illustrated in the lower part of Fig. 1(a), adding a new function to an existing workflow (the graph formed by the purple nodes) necessitates adding numerous edges (dashed lines) to enable transitions back to the "plotline" state of the workflow. The introduction of merely one out-of-workflow (OOW) response node (represented by the yellow diamond) causes dramatic increase of complexity.

On the other hand, **prompt-based** methods allow LLMs to autonomously control the dialogue process. These methods represent workflows in the text form, using natural language or code-based syntax (Fig. 1(b) shows three typical syntaxes), which are then fed into the LLM to execute actions (e.g., calling tools) or generating responses. While prompt engineering provides soft control over the LLM's behavior LLMs, being probabilistic models, still suffer from hallucinations that fail to ensure procedural *compliance* (Zhang et al., 2023).

As illustrated in Fig. 1(c), these methods occupy different points on the axis formed by the trade-off between flexibility and compliance. This leads to the central question of this paper: **How can we enhance the compliance of LLMs when performing workflow tasks without compromising their flexibility in interactions?**

Addressing this question involves two main challenges: 1) *In which form can we represent workflows precisely*? 2) *How to control the behavior of LLMs effectively*? To tackle the first challenge, we introduce a Procedure Description Language (PDL) that combines the strengths of natural language and code, allowing PDL to retain both flexibility and precision. Its adaptable syntax enables comprehensive node definitions, making it suitable for expressing various types of workflows with high representational capacity (see Sec. 4.1). In response to the second challenge, we propose the FLOWAGENT framework, which defines a series of controllers that regulate the agent's behavior based on nodes defined in the PDL. This ensures that while LLMs make autonomous decisions, they can be monitored and controlled under legal actions (see Sec. 4.2). Notably, the flexible design of PDL and the adjustable controllers within our framework allow us to balance the flexibility and compliance of FLOWAGENT (see Fig. 1(c)).

Our contributions can be summarized as follows:

1. We systematically analyze existing LLM-based workflow agents from the perspectives of compliance and flexibility. Building upon the analysis, we propose the PDL syntax for formulation of workflows, which combines the advantages of natural language and code for flexible descriptions of node relationships and overall workflow procedures.

2. We propose the FLOWAGENT framework to regulate the execution process of workflow agents. By designing different controllers, we can dynamically adjust the compliance and flexibility of workflow agents.

3. We construct an evaluation benchmark on top of existing ones to design a comprehensive evaluation method to assess the performance of workflow agents under OOW scenarios.

4. Experimental results on three datasets demonstrate that FLOWAGENT can achieve strong performance in both compliance and flexibility. Further pioneering exploration on WikiHow demonstrates our adaptability to workflow-based QA tasks.

## 2 RELATED WORK

### 2.1 LLM-BASED TASK-ORIENTED DIALOG

Unlike Open-Domain Dialog (ODD) systems, which engage in general conversation without a specific goal, Task-Oriented Dialog (TOD) systems are designed to assist users in achieving explicit, domain-specific goals, such as booking a flight or making a restaurant reservation (Yi et al., 2024; Zhang et al., 2019; Bao et al., 2019). Traditional TOD systems operate through a pipeline structure, handling tasks in distinct modules, including Natural Language Understanding (NLU), Dialogue State Tracking (DST), Dialogue Policy Management, and Natural Language Generation (NLG) (Yi et al., 2024). This modular, cascaded approach, however, often results in accumulated errors across stages (He et al., 2022; Su et al., 2021).

With recent advancements in large language models, a "workflow agent" paradigm has emerged where the LLM autonomously manages the dialogue by integrating external knowledge or using system prompts to guide interactions (Xiao et al., 2024; Zhu et al., 2024; Wallace et al., 2024). This shift has also driven changes in evaluation metrics: instead of focusing on traditional measures like the recognition accuracy of user intent and infilling slots, emphasis is now put on end-to-end task completion rates and subjective scoring of user experience (Xiao et al., 2024; Arcadinho et al., 2024). In light of this, this paper presents a new dataset and an end-to-end evaluation framework, alongside a system designed to harness and maximize the potential of LLMs in TOD scenarios.

### 2.2 LLM-BASED AGENTS AND AGENTIC WORKFLOWS

The advancement of large language model technology has fostered the development of LLM-based agents, which are now being applied across a wide range of domains (Park et al., 2023; Tang et al., 2023; Qian et al., 2023). Compared to standalone language models, LLM-based agents possess enhanced capabilities such as tool utilization, memory retention, and self-reflection, which allow for improved performance in real-life applications (Xi et al., 2023; Chu et al., 2023). Generally, these approaches aim to enhance agent performance through either model-driven planning and reasoning (Wei et al., 2022; Yao et al., 2022) or by providing external tools and knowledge sources (Schick et al., 2023; Wang et al., 2023; Zhu et al., 2024).

This evolution has introduced the concept of the *agentic workflow*, which refers to AI agents capable of autonomous planning, decision-making, and action execution to achieve goals, often without direct human intervention (Li et al., 2024; Xu et al., 2024; Liu et al., 2023; Chen et al., 2023). Notably, the Agentic Workflow approach places significant emphasis on the planning and reasoning abilities of language models, enabling them to decompose a single complex problem into a sequence of sub-tasks, or a "workflow", which serves as an intermediate state guiding the agent in subsequent steps (Valmeekam et al., 2022; Zhang et al., 2024; Xue et al., 2024). In contrast, the concept of a *workflow agent* emphasizes that, within an existing workflow, the language model follows this predefined process to accomplish tasks, often using dialogue to satisfy user needs (Xiao et al., 2024; Qiao et al., 2024). From a more general perspective, the former can serve as a logical precursor to the latter by automating the construction of the conversational workflow. However, in this paper, *we treat the workflow as pre-defined knowledge* and focus on building more robust and user-friendly agents based on these workflows.

## 3 PRELIMINARY AND BACKGROUND

### 3.1 WORKFLOW

A workflow describes the process that an agent should follow in a specific scenario or task. For example, in a hospital appointment booking scenario, the agent needs to ask the user for details such as the desired hospital, department, and preferred time, use the relevant tools to retrieve available appointment slots, confirm with the user, and complete the booking process.

Abstractly, a workflow can include operations such as asking the user for information, invoking tools, and responding to the user, which can be represented as a series of nodes. Additionally, these nodes have temporal and dependency relationships, represented as directed edges. Therefore, a

workflow can be modeled as a graph structure, specifically a directed acyclic graph (DAG), denoted by $\mathcal{G}(\mathcal{V}, \mathcal{E})$ (Qiao et al., 2024; Zhang et al., 2024).

To integrate structured workflows with language models that process linear text, two primary methods are typically used to implement workflows. The first approach, called **rule-based**, involves programming the workflow's transition rules as fixed logic, where the current node and transitions between states are hard-coded in the program. The second approach, known as **prompt-based**, represents the workflow in various formats such as natural language, code (or pseudocode), or flowchart syntax (Xiao et al., 2024; Zhu et al., 2024).

### 3.2 WORKFLOW AGENT

A workflow agent can be viewed as an agent that makes sequential decisions throughout its interactions with the user and available tools (environemnt), which can be modeled using a Markov Decision Process (MDP). The current state is denoted as $s$, the action taken by the agent as $a$, and the feedback from the environment (user responses or tool-generated outputs) as $r$. This process can be represented as $\{(s_0, a_0, r_0), (s_1, a_1, r_1), \ldots, (s_{t-1}, a_{t-1}, r_{t-1})\}$. Based on this, the decision-making process of the workflow agent can be expressed as:

$$a_t \leftarrow \mathcal{A}(\mathcal{H}_{t-1}, \mathcal{G}), \tag{1}$$

where $\mathcal{H}_{t-1}$ encompasses all actions and observations up to time $t-1$, and $\mathcal{G}$ serves as the guide for the agent's actions.

Based on the aforementioned workflow representations, workflow agents can be classified into two categories. The first category is **rule-based agents**, where the procedure in the workflow is implemented through programming. Typical examples include Coze (Coze, 2024), Dify (Dify, 2024), Flowise (Flowise, 2024), etc. Specifically, in these methods, the program rigidly controls the transitions between nodes, with the LLM acting as one of the nodes to generate user responses, predict parameters for tool calls, or assist with node transitions (e.g., classifying user intent). In such scenarios, the agent's accessible information and action space are limited, expressed as:

$$a_t \leftarrow \mathcal{M}^v(\phi^v(\mathcal{H}_{t-1}), \psi^v(\mathcal{G})), \tag{2}$$

where $v$ is the current node, $\phi^v(\mathcal{H}_{t-1})$ is the selected information visible to $v$, $\psi^v(\mathcal{G})$ is a subgraph of $\mathcal{G}$ expanded from $v$, and $\mathcal{M}^v$ denotes the language model bound to $v$. In general, this approach ensures that workflows operate under strict control but comes with high implementation costs, and predefining all state transitions is challenging to complete. For example, as shown in Fig. 2(b) session 1, when a user requests to change the hospital for an appointment, the ideal response would be for the agent to check the availability at Hospital B and inform the user. However, if the pre-configured workflow lacks an option for switching hospitals at that decision node, the rule-based workflow typically fails to respond to this need.

The second category is **prompt-based agents** (Xiao et al., 2024; Zhu et al., 2024), where a format $f$ is used to represent the workflow as linear text $\mathcal{W}^{(f)}$, and a single language model $\mathcal{M}$ autonomously manages the entire decision-making and dialogue process. This process can be represented as:

$$a_t \leftarrow \mathcal{M}(\mathcal{H}_{t-1}, \mathcal{G}^{(f)}), \tag{3}$$

where $\mathcal{G}^{(f)}$ is the graph structure implicitly conveyed in $\mathcal{W}^{(f)}$. Compared to rule-based methods, the LLM in this approach can access the complete conversation history and workflow information, and its action space encompasses all nodes on $\mathcal{G}$. Thus, these methods allow the LLM to autonomously determine state transitions within the workflow, offering greater flexibility. However, since the LLM is inherently a probabilistic model, it is prone to making errors, making it difficult to ensure procedural compliance. However, since the LLM is inherently a probabilistic model, it is prone to making errors, making it difficult to ensure procedure compliance. Additionally, although some preliminary exploration has been conducted, the impact of different workflow representation syntaxes on the performance of LLMs as agents has not been fully studied (Xiao et al., 2024).

## 4 METHOD

In this paper, we propose a new procedural description language, PDL, to represent workflows, and implement an execution framework, FLOWAGENT, to control the agent's behavior. Considering the characteristics of LLMs, PDL is designed to represent workflows using a mix of natural

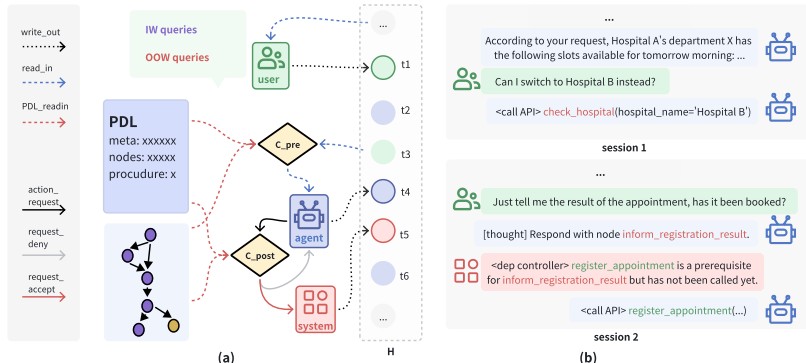

Figure 2: Subfigure (a) shows the data and control flow of FLOWAGENT. Subfigure (b) displays two sessions of FLOWAGENT under the hospital appointment workflow.

language and code, combining the flexibility of natural language and precision of code language. Besides, sparse dependencies expressed in PDL ensure that LLMs can autonomously decide how to proceed alongside the conversation with user. The syntax of PDL will be discussed in Sec. 4.1. Furthermore, to enhance the compliance of the workflow agent, we have developed an execution framework FLOWAGENT that works in conjunction with PDL. It applies a series of controllers during the agent's decision-making process, enabling reliable action execution while still allowing the language model to make autonomous decisions. Further details about FLOWAGENT can be found in Sec. 4.2.

## 4.1 PDL SYNTAX

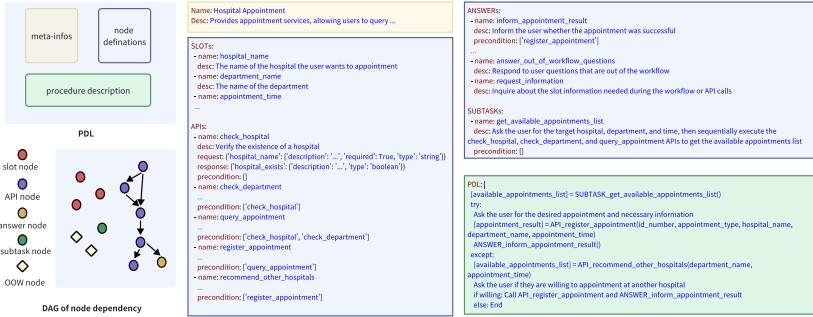

Figure 3: PDL example in the hospital appointment scenario. The middle and right parts show the three components in PDL syntax: meta information, node definitions, and procedure description. The lower left corner of the figure displays the DAG formed by the node dependencies of this PDL.

We propose a new procedure description language (PDL) that integrates natural language and code to represent workflows. As shown in Fig. 3, the PDL adopts a YAML-like syntax and consists of three main parts: *meta information, node definitions, and procedure description*: $\mathcal{W}^{(pdl)} = \{I, N, P\}$, where meta information $I$ contains basic information about the workflow (e.g., name, description), node definitions $N$ defines the resources the agent may use during execution (e.g., tools, response strategies), and procedure description $P$ details the procedure that needs to be followed for this task, expressed using a mix of natural language and pseudocode.

The node definitions in the PDL syntax provide essential information for the agent's execution process and define the agent's available action space. We have designed three types of nodes: **SLOT nodes**, which represent necessary slots during workflow execution or API calls, such as hospital_name and department_name in a hospital registration workflow; **API nodes**, which define external tools that the agent can call, including functions like check_hospital, check_department, query_appointment, and register_appointment; and **ANSWER nodes**, which specify the agent's response behavior, such as inform_registration_result, directing the agent's response after a registration attempt. Additionally, an answer_OOW_questions node allows the LLM to flexibly handle user

queries that fall outside the defined workflow, addressing limitations commonly found in rule-based workflow agents.

It is important to note that node definitions include a *preconditions* attribute, which outlines dependencies between nodes. For instance, the `register_appointment` node lists `['query_appointment']` as a prerequisite, indicating that the agent must query a suitable appointment list before proceeding with registration to prevent invalid actions. Similarly, the `inform_registration_result` node must follow `['register_appointment']`, ensuring that responses are generated only after a registration attempt has been completed, thus avoiding false responses caused by LLM hallucinations. By leveraging these *preconditions*, a node dependency graph $\mathcal{G}^{(pdl)}$ can be reconstructed, similar to the $\mathcal{G}$ discussed in Sec. 3, but potentially differing in node definitions and topology.

In summary, PDL offers three main features:
1. **Simplicity and flexibility**: PDL allows for summarizing essential tools and actions within a workflow through nodes, covering common slot descriptions, API calls, and response nodes. The syntax is user-friendly, enabling users to quickly create PDL workflows tailored to their scenarios without the steep learning curve of strict code language or flowchart syntax.
2. **Structured yet flexible representation**: The structured expression, enhanced with natural language descriptions, makes the workflow representation concise while retaining the natural language's flexibility and the preciseness of code.
3. **Explicit dependency representation**: PDL can accurately define logical relationships between nodes, similar to rule-based methods that implement abstract workflow graphs $\mathcal{G}$ but without using rigid "If...Then..." conditions. Instead, it enforces legal or illegal transitions through sparse dependencies, allowing LLMs to autonomously decide node transitions, thus ensuring the agent's flexibility.

## 4.2 FLOWAGENT ARCHITECTURE

To complement the PDL syntax, we developed an execution framework that applies controllers during the agent's decision-making process, ensuring reliability while allowing autonomous decision-making by the language model.

As illustrated in Fig. 2(a), the framework models the system using a multi-agent interaction structure with three main roles that share a conversation history $\mathcal{H}$ (shown in the dashed box, where messages flow from top to bottom over time). (1) **User** ($\mathcal{U}$): Engages in the conversation by expressing needs, which may be related to the workflow (in-workflow, IW) or outside the workflow (out-of-workflow, OOW). (2) **Workflow Agent** ($\mathcal{A}$): Understands the user's needs and generates corresponding responses or actions. (3) **System** ($\mathcal{S}$): Offers tools that the agent can call upon.

The overall execution process of FLOWAGENT is detailed in Alg. 1. During each interaction round, the user poses a new query (line 3), and the agent interprets the user's intent, potentially calling a tool (line 18) and ultimately producing a response for the user (line 21).

To ensure stable decision-making by the agent, we incorporated two groups of controllers: pre-decision controllers ($\mathcal{C}_{\text{pre}} = \{c_i^{\text{pre}}\}_{i=1}^{C_{\text{pre}}}$) and post-decision controllers ($\mathcal{C}_{\text{post}} = \{c_j^{\text{post}}\}_{j=1}^{C_{\text{post}}}$). **Pre-decision controllers** guide the agent's actions before decisions are made. Each controller $c_i^{\text{pre}}$ generates a textual evaluation result $r_i$ based on the current state (e.g., determining which nodes are invalid according to the current state of $\mathcal{G}^{(pdl)}$). These results, $\mathcal{R}_{\text{pre}}$, are provided as input to the language model, serving as a soft form of control. While pre-decision controllers help guide the LLM's behavior, the LLM may still produce unstable outputs. Therefore, **post-decision controllers** assess the validity of the agent's actions after they are generated, serving as a hard constraint. In Fig. 2(a), the agent's output $\mathcal{O}_{\mathcal{A}}$ is considered an "action_request". Each post-decision controller $c_j^{\text{post}}$ evaluates the legitimacy of this request, resulting in either "request_deny" or "request_accept".

Specifically, we designed various modular controllers to adjust the behavior of the workflow agent across multiple dimensions, such as enforcing node dependencies, constraining API call repetition, limiting conversation length, and preventing hallucinations in LLM outputs. Below, using the workflow shown in Fig. 3 as an example, we briefly introduce two controllers based on node dependencies and API call repetition:

1. **Node Dependency Controllers**: These controllers can function as either pre-decision or post-decision mechanisms. When acting as a pre-decision controller, $c_{\text{dep}}^{\text{pre}}$ evaluates the agent's current node and retrieves a list of inaccessible nodes by validating conditions on the dependency graph. For instance, if the agent is at the `check_hospital` node, $c_{\text{dep}}^{\text{pre}}$ identifies that the `query_appointment` node is unreachable due to unmet prerequisites in $\mathcal{G}^{(pdl)}$, preventing the LLM from jumping to that node and thus enabling soft control. As a post-decision controller, $c_{\text{dep}}^{\text{post}}$ checks the legitimacy of node transition requests. For example, if the agent attempts to move to `query_appointment` without completing `check_department`, the controller identifies the violation of dependencies and denies the request, sending feedback to the agent.

2. **API Call Repetition Controllers**: These controllers track the agent's API call history to prevent repetitive calls with the same parameters, a common issue in language models. This can also function in both pre-decision and post-decision modes. As a pre-decision controller, $c_{\text{api}}^{\text{pre}}$ identifies APIs that have reached their call limits and removes them from the list of available tools. For example, if the `check_hospital` API has been called twice for *Hospital A* with negative results, the controller excludes it to prevent further redundant calls. In post-decision mode, $c_{\text{api}}^{\text{post}}$ enforces stricter constraints; if the agent attempts another call to `check_hospital`, the controller intervenes and sends a message to the agent: "API call denied: `check_hospital` has reached its limit. Please use a different tool."

In summary, pre-decision controllers act as "guides" by refining the agent's action space before decisions are made, while post-decision controllers function as "gatekeepers" that validate the legitimacy of the agent's outputs.

# 5 EVALUATION AND DATA

## 5.1 EVALUATION METHOD

We follow previous studies (Xiao et al., 2024; Chen et al., 2023) to conduct both turn-level and session-level assessments for evaluation of the proposed framework.

**Turn-level evaluation** Similar to the classic TOD task evaluation (Dai et al., 2022), there is a reference session (considered as ground truth). For each turn in the reference session, the evaluation system provides the prefix of the session $\mathcal{H}_{t-1}$ to the bot for predicting the current $\hat{a}_t$. The judge compares $\hat{a}_t$ with $a_t$ to determine if the bot's response for that turn is correct, and the average result across all turns yields the *Pass Rate*. Unlike Xiao et al. (2024) that adopts powerful LLM such as GPT4 as a judge for scoring the generated contents of the LLM agent, we do not introduce LLM-based scoring because we believe the evaluation is prone to be biased to the preference of the judging LLM for certain styles, lengths, and formats rather than contents. Instead, we use a binary classification of correctness and calculate the pass rate based on this. To assess the agent's tool usage capability, for turns involving tool callings, we evaluate the tool selection and parameter infilling performance of the agent in *Precision, Recall, and F1-score*.

**Session-level evaluation** In session-level evaluation, considering the cost of assessment, we use an LLM to simulate the user and interact with the bot. To ensure these simulated conversation sessions closely resemble real-world scenarios rather than simplistic examples, we define a user profile for each user, including: (1) basic demographic information; (2) conversational style detailing the user's behavior patterns; and (3) user needs for the workflow, describing the main goals for the session (which may include some secondary objectives). A specific example of a user profile can be found in App. A.2. For each generated session, we conduct a binary evaluation to determine whether the user's primary intention for the workflow is achieved, resulting in the *Success Rate* metric. Additionally, we track the number of sub-tasks expressed by the user and the number completed within the session to derive the *Task Progress* metric. We use the same prompts as proposed in Xiao et al. (2024) to evaluate each session in an end-to-end manner. We also assess the tool-calling performance of the LLM agent using *Precision, Recall*, and *F1-score*.

## 5.2 FLEXIBILITY EVALUATION

Previous work (Zhong et al., 2018; Wu et al., 2019; Li et al., 2024) has primarily focused on evaluating whether bots can follow a specific procedure to complete a conversation, which partially emphasizes compliance while neglecting flexibility in handling user requests. Such incomprehensive evaluation may not reflect the capabilities of LLM agents under real-world scenarios, where an "imperfect" user might not adhere to the procedure and violates the sequential steps during multiple rounds of interactions. Consequently, to evaluate the performance of workflow agents in OOW scenarios, we have additionally developed a targeted evaluation method to assess flexibility.

Specifically, we categorize OOW scenarios into three types: (1) *intent switching*, where the user suddenly changes the original intent requests or requirements, including modification of API slots/parameters and demand for cancellations; (2) *procedure jumping*, where the user does not follow the established workflow sequence to provide information and express confirmation, including skipping steps or jumping back; and (3) *irrelevant answering*, where the user deliberately avoids direct reply to questions raised by the agent, such as answers with topic shifts and rhetorical questions;

Given the defined categories above, we can evaluate the flexibility of the agent by observing its performance in OOW scenarios. We adopt the metrics defined in Sec. 5.1. At the turn-level, we evaluate the agent's immediate flexibility by inserting OOW user queries and observing its performance in these specific turns. At the session-level, we assess the agent's overall performance in sessions that include OOW queries to measure its long-term flexibility.

## 5.3 DATA

We constructed three test datasets based on existing datasets and business-related data: SGD (Rastogi et al., 2019), STAR (Mosig et al., 2020), and In-house. The data construction process is detailed in App. D.2. Statistics for these datasets are shown in Tab. 4, and differences from datasets used in other studies are highlighted in Tab. 5.

Specifically, our datasets include: (1) four types of workflows (see App. A); (2) user profiles required for session-level evaluation (see App. A.2); and (3) conversations needed for turn-level evaluation (see App. B.1).

## 6 EXPERIMENTS

To extensively measure the performance of LLM agents in delivering automatic workflow handling capabilities, we raise the following research questions:

**Q1**: Compared with other models, does our proposed FLOWAGENT show improvements in compliance and flexibility?
**Q2**: In which way the proposed controllers exert constraints on the model to facilitate workflows with both compliance and flexibility?
**Q3**: Can the proposed FLOWAGENT framework be applied to a wider range of scenarios?

## 6.1 EXPERIMENTAL SETUP

**Baselines**: We selected ReAct (Yao et al., 2022) as a baseline method for comparison, which makes decisions in each round by utilizing a combination of *thought* and *action*, and treats the feedback from environment an *observation*. It belongs to the category of prompt-based methods introduced in Sec. 3.2. For representing the workflow, we chose three formats: natural language (NL), code, and FlowChart, denoted as $ReAct_{NL}$, $ReAct_{code}$, and $ReAct_{FC}$, respectively. To ensure a fair comparison, we reused the prompts from FlowBench (Xiao et al., 2024) in our experiments.

**Implementation**: In session-level evaluation, `GPT-4o-mini` is used for user and system simulation. For the bot, we initially tested two representative model series, the GPT series (Achiam et al., 2023) and the Qwen series (Yang et al., 2024), which include both open-source and proprietary models. Preliminary studies revealed that small, weak models are not competent for complex workflow tasks. Therefore, in the present study, we choose `GPT-4o` and `Qwen2-72B` for demonstrations.

During the evaluation process, we used `GPT-4-Turbo` for judgment. More implementation details can be seen in App. C.1.

## 6.2 SESSION-LEVEL EXPERIMENTAL RESULTS

Table 1: Session-level Evaluation Results

| Backbone Model | Method | In-house dataset | | | STAR | | | SGD | | |
|---|---|---|---|---|---|---|---|---|---|---|
| | | Success Rate | Task Progress | Tool F1 | Success Rate | Task Progress | Tool F1 | Success Rate | Task Progress | Tool F1 |
| GPT-4o | ReAct$_{NL}$ | 62.50 | 80.33 | 63.16 | 40.17 | 78.33 | 76.96 | **34.62** | 82.44 | **89.11** |
| | ReAct$_{code}$ | 57.26 | 75.20 | 75.86 | 38.27 | 75.10 | 55.32 | 29.23 | 76.67 | 82.32 |
| | ReAct$_{FC}$ | 60.01 | 82.70 | 72.00 | 33.43 | 72.58 | 82.33 | 30.92 | 81.24 | 85.71 |
| | FLOWAGENT | **67.72** | **85.12** | **80.60** | **42.78** | **80.42** | **84.00** | 32.79 | **84.21** | 86.60 |
| Qwen2-72B | ReAct$_{NL}$ | 40.51 | 80.01 | 78.90 | 16.67 | 59.34 | 82.12 | 13.46 | 67.94 | 84.42 |
| | ReAct$_{code}$ | 32.78 | 65.58 | 75.20 | 10.42 | 56.70 | 63.63 | 15.76 | 59.84 | 72.55 |
| | ReAct$_{FC}$ | 41.67 | 80.97 | 77.78 | 9.21 | 53.80 | 61.58 | 28.79 | 62.98 | 85.40 |
| | FLOWAGENT | **44.32** | **82.22** | **84.21** | **18.42** | **61.42** | **86.86** | **30.84** | **69.91** | **88.02** |

Table 2: Session-level Evaluation Results in OOW Scenarios

| Backbone Model | Method | In-house dataset | | | STAR | | | SGD | | |
|---|---|---|---|---|---|---|---|---|---|---|
| | | Success Rate | Task Progress | Tool F1 | Success Rate | Task Progress | Tool F1 | Success Rate | Task Progress | Tool F1 |
| GPT-4o | ReAct$_{NL}$ | 18.03 | 72.20 | 75.42 | 4.55 | 43.59 | 81.58 | 3.31 | 49.42 | 74.12 |
| | ReAct$_{code}$ | 16.23 | 57.27 | 73.68 | 2.08 | 40.74 | 70.21 | 2.92 | 54.23 | 64.57 |
| | ReAct$_{FC}$ | 18.21 | 71.42 | 78.57 | 5.17 | 43.52 | 82.05 | 4.02 | 47.57 | 73.56 |
| | FLOWAGENT | **32.01** | **75.20** | **81.57** | **10.21** | **52.31** | **85.32** | **7.16** | **56.64** | **77.83** |
| Qwen2-72B | ReAct$_{NL}$ | 16.76 | 69.41 | 72.27 | 6.25 | 48.30 | 82.92 | 5.01 | 47.00 | 82.83 |
| | ReAct$_{code}$ | 0.00 | 60.41 | 71.62 | 2.02 | 45.31 | 70.80 | 2.08 | 45.35 | 70.79 |
| | ReAct$_{FC}$ | 17.14 | 70.42 | 75.56 | 0.00 | 45.63 | 84.49 | 4.10 | 46.33 | 78.29 |
| | FLOWAGENT | **30.20** | **75.70** | **80.01** | **8.72** | **50.28** | **86.72** | **8.25** | **49.30** | **89.88** |

**A1.1: FLOWAGENT outperforms the other three baselines in terms of task compliance.** We first compare the session-level performance of different methods in Tab. 1. The results indicate that FLOWAGENT outperforms the other three baselines in terms of task completion metrics *Success Rate*, *Task Progress*, and tool usage metrics like *Tool F1*.

**A1.2: FLOWAGENT exhibits robustness towards OOW interventions with higher flexibility.** Tab. 2 presents the performance of different methods under OOW scenarios. A general performance decline is observed across all models on the three datasets. However, FLOWAGENT exhibits only a slight decline, achieving the best results across all datasets. Fig. 5(a) visualizes the *Task Progress* metric under different settings, highlighting FLOWAGENT's advantage in OOW scenarios, demonstrating strong flexibility.

## 6.3 TURN-LEVEL EXPERIMENTAL RESULTS

**A1.3: FLOWAGENT maintains the superior compliance and flexibility across datasets in turn-level evaluation.** We present the turn-level experimental results of Qwen2-72B in Tab. 6. The results show that the FLOWAGENT framework achieves the best performance in both IW and OOW settings. What's more, Fig. 5(b) compares the *Success Rate* across different models and settings.

## 6.4 ABLATION STUDIES

Table 3: Ablation Study Results

| Method | In-house dataset | | | STAR | | | SGD | | |
|---|---|---|---|---|---|---|---|---|---|
| | Success Rate | Task Progress | Tool F1 | Success Rate | Task Progress | Tool F1 | Success Rate | Task Progress | Tool F1 |
| FLOWAGENT | **57.26** | **84.71** | 76.13 | **22.22** | 70.44 | **91.89** | **16.67** | **69.89** | 89.89 |
| -post | 55.71 | 84.56 | **76.70** | 20.83 | **72.57** | 90.20 | 8.33 | 66.28 | 83.98 |
| -post-pre | 43.75 | 80.50 | 75.00 | 12.50 | 63.75 | 86.27 | 7.69 | 65.77 | 88.66 |

**A2: Controllers play an indispensable role in enforcing steady progress of workflows with OOW interventions.** We conducted ablation experiments on FLOWAGENT in OOW settings, with the results shown in Tab. 3. In the table, "-post" indicates the removal of the post-decision controllers $\mathcal{C}_{\text{post}}$ from the complete model, while "-post-pre" further removes the pre-decision controllers $\mathcal{C}_{\text{pre}}$. According to the experimental results, it is evident that removing either controller negatively impacts model performance, validating that controllers in FLOWAGENT enhance the model's compliance.

## 6.5 EXPLORATORY EXPERIMENT ON WIKIHOW

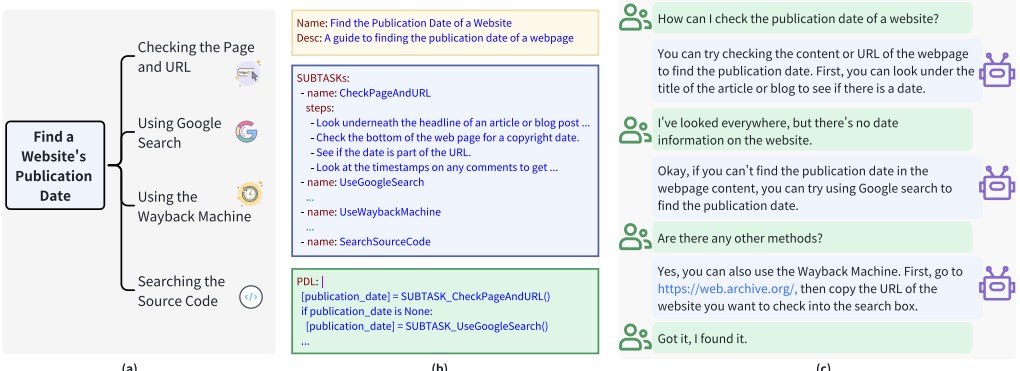

Figure 4: WikiHow data and workflow-based QA. Subfigure (a) shows the content on the page https://www.wikihow.com/Find-the-Publication-Date-of-a-Website; Subfigure (b) shows the PDL we derived from it (see App. A.3); Subfigure (c) shows an example conversation between FLOWAGENT and a user.

In practical applications, we found that besides using tools to help users complete tasks, there is another category of tasks involving procedural descriptions. In these cases, the bot does not actively call tools; instead, the user performs the necessary actions. For example, a user uploads an appliance manual and engages in QA with the bot to learn how to perform repairs. We define these tasks as **workflow-based QA**. We converted workflows from the WikiHow website into PDL syntax, covering four categories with 20 examples, to evaluate whether PDL and FlowAgent are suitable for this task. (For more on the task background and data construction, see Appendix E.3.)

**A3: The FLOWAGENT framework shows strong potential for broad applicability to real-world workflow-based QA tasks.** We tested this through manual interactions and found that PDL syntax effectively represents WikiHow-like workflows, and the FLOWAGENT framework supports this new task. Fig. 4(ab) display a WikiHow workflow and its PDL format, while Figure4(c) shows a sample dialogue based on this workflow. (For more detailed examples, see App. A.3 and B.) Future work includes developing a standard benchmark and an interactive evaluation environment for this task.

## 7 CONCLUSION

In this paper, we reviewed existing LLM-based workflow methods and compared their strengths and weaknesses in terms of compliance and flexibility. Aiming to enhance the compliance capability of LLMs without significantly compromising their flexibility, we proposed the PDL syntax to express workflows and used the FLOWAGENT framework to control agent behavior. For evaluating compliance and flexibility capabilities, we constructed datasets based on existing data and designed specific evaluation methods. Experiments on three datasets demonstrated that FLOWA-GENT not only possesses strong compliance capabilities but also exhibits robust flexibility when handling OOW queries. Additionally, we validated FLOWAGENT's potential in workflow-based QA tasks using the WikiHow dataset, inspiring our future research.

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

## A  DATASET EXAMPLES

### A.1  PDL EXAMPLE

Below is a PDL example in a real-world scenario. For formats of natural language, code and flowchat, see Xiao et al. (2024).

```
Name: 114 Registration
Desc: Provides registration services, allowing users to query and
recommend hospitals and departments in Beijing.
Detailed_desc: Queries the availability of appointment slots based on the
 user's specified hospital, department, and time, and attempts to
register; if no slots are available at the specified hospital, it will
try to register at other hospitals.

SLOTs:
  - name: hospital_name
    desc: The name of the hospital where the user wants to register.
  - name: department_name
    desc: The name of the department where the user wants to register.
  - name: appointment_time
    desc: The time when the user wants to register.
  - name: id_number
    desc: The user's ID number.
  - name: registration_type
    desc: The type of registration (specialist or general).
  - name: doctor_name
    desc: The name of the doctor the user wants to register with.
  - name: registration_willingness
    desc: Whether the user is willing to register at other hospitals.
  - name: registration_status
    desc: The result of the registration as returned by the API, where 1
    indicates success and 0 indicates failure.

APIs:
  - name: check_hospital
    request: [hospital_name]
```

```
      response: [hospital_exists]
      precondition: []
   - name: check_department
      request: [department_name, hospital_name]
      response: [department_exists]
      precondition: [check_hospital]
   - name: query_appointment
      request: [hospital_name, department_name, appointment_time]
      response: [available_slots, available_list, specialist_count,
      general_count]
      precondition: [check_hospital, check_department]
   - name: recommend_other_hospitals
      desc: Searches for available slots at other hospitals for the
      specified department and time.
      request: [department_name, appointment_time]
      response: [available_slots, available_list]
      precondition: [check_department]
   - name: register_hospital
      request: [id_number, registration_type, hospital_name,
      department_name, appointment_time]
      response: [registration_status]
      precondition: [query_appointment]
   - name: register_other_hospital
      request: [id_number, hospital_name, doctor_name]
      response: [registration_status]
      precondition: [recommend_other_hospitals]

ANSWERs:
   - name: Hospital not found
      desc: Sorry, we currently cannot provide registration services for
      this hospital. Please contact the hospital directly or consider other
       hospitals.
   - name: Department not found
      desc: $hospital_name does not have the department you are looking for
      . I will transfer you to a customer service representative for
      further assistance. Please wait.
   - name: No available slots
      desc: We apologize, but there are no available slots for the
      department you want to register at any hospital on our platform.
      Please follow the WeChat public account "Beijing 114 Appointment
      Registration" to register as per your needs. Thank you for calling,
      and have a nice day.
   - name: Registration refused
      desc: Please follow the WeChat public account "Beijing 114
      Appointment Registration" to register as per your needs. Thank you
      for calling, and have a nice day.
   - name: Hospital registration successful
      desc: Your registration at $hospital_name $department_name for
      $appointment_time has been successful. A confirmation message will be
       sent to your phone number shortly. Is there anything else I can help
       you with?
   - name: Hospital registration failed
      desc: We apologize, but there are no available $registration_type
      slots at $hospital_name $department_name for $appointment_time.
      Please follow the WeChat public account "Beijing 114 Appointment
      Registration" to register as per your needs. Thank you for calling,
      and have a nice day.
   - name: Other hospital registration successful
      desc: Your registration at $recommend_other_hospitals-hospital_name
      with $recommend_other_hospitals-doctor_name for $appointment_time has
       been successful. A confirmation message will be sent to your phone
      number shortly. Is there anything else I can help you with?
   - name: Other hospital registration failed
      desc: We apologize, but the ID information is incorrect, and we
      cannot proceed with the registration. Please follow the WeChat public
```

```
      account "Beijing 114 Appointment Registration" to register as per
    your needs. Thank you for calling, and have a nice day.
  - name: Other free response questions
  - name: Please provide necessary information

PDL: |
  [hospital_exists] = API_check_hospital([hospital_name])
  if hospital_exists == false:
      ANSWER_Hospital_not_found()
  elif hospital_exists == true:
      [department_exists] = API_check_department([department_name,
      hospital_name])
      if department_exists == false:
          ANSWER_Department_not_found()
      elif department_exists == true:
          [available_slots, available_list, specialist_count,
          general_count] = API_query_appointment([hospital_name,
          department_name, appointment_time])
          if available_slots > 0:
              [registration_status] = API_register_hospital([id_number,
              registration_type, hospital_name, department_name,
              appointment_time])
              if registration_status == "1":
                  ANSWER_Hospital_registration_successful()
              elif registration_status == "0":
                  ANSWER_Hospital_registration_failed()
          elif available_slots == 0:
              [available_slots, available_list] =
              API_recommend_other_hospitals([department_name,
              appointment_time])
              if available_slots > 0:
                  if registration_willingness == "true":
                      [registration_status] = API_register_other_hospital
                      ([id_number, hospital_name, doctor_name])
                      if registration_status == "1":
                          ANSWER_Other_hospital_registration_successful()
                      elif registration_status == "0":
                          ANSWER_Other_hospital_registration_failed()
                  elif registration_willingness == "false":
                      ANSWER_Registration_refused()
              elif available_slots == 0:
                  ANSWER_No_available_slots()
```

Listing 1: Example of PDL

## A.2 USER PROFILE EXAMPLE

Below is an example of a used user profile. The "User Details" contains some randomly generated attributes; "Dialogue Style" specifies the user's conversational style; "User Needs" describes the user's requirements related to a specific workflow; "Interactive Pattern" further details the possible dialogue process for the user within that workflow.

```
**Persona**:
A 25-year-old bartender with three years of experience in the hospitality
 industry. He is known for his honesty, often giving customers sincere
advice on their drink choices.

**User Details**:
- Name: Michael James Carter
- Sex: Male
- Age: 25
- Phone Number: 13812345678
- ID Number: 11010519980101 2345
```

```
**User Needs**:
- Michael needs to query available appointment slots for specific
hospitals and departments in Beijing.
- He may need to verify the existence of certain hospitals and
departments.
- He wants to make an appointment for a medical consultation at a
preferred hospital and department.
- If the preferred hospital or department is not available, he may need
recommendations for alternative hospitals and departments.
- Michael may also need to know the success or failure status of his
appointment registration.

**Dialogue Style**:
- Michael's dialogue style is likely to be straightforward and sincere,
reflecting his honesty in his profession as a bartender.
- He may prefer clear and concise information without unnecessary jargon.
- His tone is likely to be polite and respectful but also direct, as he
is used to providing sincere advice to customers.
- He may appreciate a friendly and helpful attitude from the assistant.

**Interactive Pattern**:
- Michael might start by specifying the hospital and department he is
interested in.
- He is likely to ask for available appointment slots for a specific time
.
- If the hospital or department does not exist, he will appreciate being
notified promptly and clearly.
- If there are no available slots at his preferred hospital, he may ask
for recommendations for other hospitals.
- He will likely ask for the success status of his appointment
registration and may need guidance on the next steps if the registration
fails.
- Michael may interact in a sequential manner, addressing one query at a
time, and expecting clear and immediate responses from the assistant.
```

Listing 2: Example of user profile

## A.3 WIKIHOW-BASED PDL EXAMPLE

Below is the PDL derived from the page https://www.wikihow.com/Find-the-Publication-Date-of-a-Website.

```
Name: Find the Publication Date of a Website
Desc: A guide to finding the publication date of a webpage

SUBTASKs:
  - name: CheckPageAndURL
    steps:
      - Look underneath the headline of an article or blog post for the
      publication date.
      - Check the bottom of the web page for a copyright date.
      - See if the date is part of the URL.
      - Look at the timestamps on any comments to get an estimate of the
      publication date.
  - name: UseGoogleSearch
    ...
  - name: UseWaybackMachine
    ...
  - name: SearchSourceCode
    steps:
      - Right-click the web page and select `View Page Source`.
      - Press `Ctrl+F` (Windows) or `Command+F` (Mac).
      - Search for the term "date" or "published".
      - Look for the date listed in year-month-day order.
```

```
PDL: |
  [publication_date] = SUBTASK_CheckPageAndURL()
  if publication_date is None:
    [publication_date] = SUBTASK_UseGoogleSearch()
  if publication_date is None:
    [publication_date] = SUBTASK_UseWaybackMachine()
  if publication_date is None:
    [publication_date] = SUBTASK_SearchSourceCode()
```

Listing 3: Example of PDL from WikiHow

# B CONVERSATION EXAMPLES

## B.1 CONVERSATION EXAMPLE ON STAR DATASET

Below is a conversation generated on the STAR dataset (for turn-level evaluation). Note that in line 16, the user simulated by the LLM adopted an "intent-switching/detail-switching" OOW and modified the appointment time.

```
1  ...
2  BOT: Great, Alex. What's the name of the apartment you're interested in?
3  USER: Maple Apartments.
4  BOT: When would you like to view the Maple Apartments?
5  USER: I'd like to do it on Friday.
6  BOT: What time on Friday would you prefer for the viewing?
7  USER: How about 3 PM?
8  BOT: Have you paid the application fee?
9  USER: Yes, I have.
10 BOT: Would you like to add any custom message for the viewing?
11 USER: No, that's fine.
12 BOT: <Call API> book_apartment_viewing({'RenterName': 'Alex', 'Name': '
   Maple Apartments', 'Day': 'Friday', 'StartTimeHour': '15', '
   ApplicationFeePaid': 'Yes', 'Message': '', 'RequestType': '
   CheckAvailability'})
13 SYSTEM: {'Status': 'Available'}
14 BOT: The viewing is available. Would you like to proceed with booking?
15 USER: Actually, I might need to change the time. Can we do 4 PM instead?
16     (OOW type) intent_switching/detail-switching
17 BOT: <Call API> book_apartment_viewing({'RenterName': 'Alex', 'Name': '
   Maple Apartments', 'Day': 'Friday', 'StartTimeHour': '16', '
   ApplicationFeePaid': 'Yes', 'Message': '', 'RequestType': '
   CheckAvailability'})
18 SYSTEM: {'Status': 'Available'}
19 BOT: The new time at 4 PM is also available. Shall I book it for you?
20 USER: Yes, please book it.
```

Listing 4: Example conversation on STAR

## B.2 CONVERSATION EXAMPLE ON WIKIHOW

An example conversation based on the WikiHow PDL shown in App A.3.

```
1  U: How to find a website's publication date using code?
2  B: To find a website's publication date, you can inspect the page source
   and search for terms like "date" or "published" to locate the date in the
    metadata.
3  U: How to inspect the page source?
4  B: To inspect the page source, you can right-click the webpage and select
    "View Page Source" in your web browser.
5  U: I got "<meta property="article:published_time" content="2024-11-20T17
   :45:37.542Z"/><meta property="article:modified_time" content="2024-11-20
```

```
T17:45:37.542Z"/><meta property="article:author" content="Ronan Farrow
"/>"
B: Great! The '<meta>' tags you found contain the publication and
modification dates for the article. The 'article:published_time' shows
when the article was first published: 2024-11-20.
...
```

Listing 5: Example conversation on WikiHow

## C  EXPERIMENTAL DETAILS

### C.1  IMPLEMENTATION DETAILS

For the GPT series, we specifically used the models `gpt-4o-2024-05-13`, `gpt-4o-mini-2024-07-18`, and `gpt-4-turbo-2024-04-09`. To ensure stable output results, we set the temperature to 0.2. For the Qwen2 series models, we utilized the vllm framework for inference, also setting the temperature to 0.2.

### C.2  PROMPTS

**Prompts for User Simulation**  Below are the prompts we used with LLM to simulate user behavior. The "User Profile" refers to Sec. A.2.

Note that for OOW simulation, we add an "additional constraints" field in the user profile to describe the user's current OOW intention. For example, "In this round, you can ask a question unrelated to the current topic" will be injected in the prompt if an "irrelevant answering" OOW intent is randomly chosen in the 5th round of dialogue. We dynamically adjust OOW queries during the conversation by controlling the injected instructions.

```
You are a real-life user that interact with an assistant of {{
assistant_description }} to achieve your specific objectives.

## User Profile
```
{{ user_profile }}
```

## History conversation
```
{{ history_conversation }}
```

## Specific requirements
1. Role Awareness: Remember you are playing the user role and speak in
the first person.
2. Goal-Oriented: Keep the conversation focused on achieving your needs.
3. Style: Keep your response concise and real-life.
4. Engagement: Maintain an engaging and curious tone to facilitate
effective dialogue.
5. Your output format should be:
```
Response: xxx (the response content)
```
6. Stop: End the conversation when the task is completed or when it
becomes repetitive and no longer meaningful to continue. Set your
response as "[END]" to stop the conversation.
```

Listing 6: Prompt for user simulation

**Inference Prompt for FLOWAGENT**  Below is the inference prompt for our FLOWAGENT.

```
You are a bot designed to assist the user for a specific task described
by the Procedure Description Language (PDL). Your goal is to engage in a
friendly conversation with the user while helping them complete the task.

### Constraints
1. **Step Identification**: Throughout the conversation, you should
determine the user's current step, (whether it is in the PDL or just
general questions), and dynamically follow PDL:
    - If the user's query aligns with the PDL logic, proceed to the next
    step.
    - If the user ask irrelevant questions, generate a response that
    maintains a fluent and logical conversation.
2. **PDL Components**: The PDL includes several components:
    - meta information: `name, desc, desc_detail` are meta information
    about the PDL.
    - slots: `slots`s define the information you may need to collect from
     user, or the values returned by the API.
    - reference answer: `answers` define the responses you should
    response to the user.
    - procedure: the final `procedure` string is a Pythonic language that
     defines the core logic of the procedure.
3. Notes:
    - You have to collect enough parameter values from the user before
    calling the apis.

### PDL
```PDL
{{ PDL }}
```

### Available APIs
{{ api_infos }}

### History Conversation
{{ conversation }}

### Current state
{{ current_state | trim }}

### Output Format
Your output format should be chosen from one of the two templates below.
1. If you need to interact with the user without calling an API (inquire
slot values or reply/answer):
```
Thought: xxx (description of your thought process )
Response: xxx (the content you need to inquire or reply)
```
2. If you need to call an API:
```
Thought: xxx (description of your thought process )
Action: xxx (the function name to be called, do not prefix "API_".)
Action Input: xxx (the parameters for the function, must be in strictly
valid JSON format)
```
```

Listing 7: Prompt for FLOWAGENT

**Inference Prompt for ReAct**   For the baseline ReAct, we directly borrowed the prompt used in
FlowBench (Xiao et al., 2024).

```
You are a helpful assistant for the task of {{task_description}}.

### Specific requirements
```

```
1. You need to act as an assistant and engage in a conversation with the
user, following the business process and API information.
2. You have been provided with the flowchart information for different
scenarios under a specific role.
3. You can only answer questions within the scope of the given several
workflow processes. If the user asks a question beyond these scopes,
please apologize and explain to the user in the response part.
4. When asking for API input parameters, ensure that the provided
parameter values comply with the specified format regarding both the
correctness of the format and the completeness of the content. Do not
assign values arbitrarily. In instances where the parameters do not meet
the format requirements, notify users to make the adjustments until the
requirements are satisfied.
5. When the user has multiple requests at the same time, please select
one appropriate request for processing first and inform the user that
other requests will be resolved subsequently. If there is unfinished
business in the previous conversation, continue to provide the necessary
help and guidance to assist them in completing the business process. When
 multiple APIs need to be called, do so in separate rounds, with a
maximum of one API call output per round. When the user indicates that
the business is finished or says goodbye, respond politely and end the
conversation.

### Workflow information
```
{{workflow}}
```

### Tool information
{{toolbox}}

### Current time
{{current_time}}

### History conversation
{{history_conversation}}

### Output format
Your output format should be chosen from one of the two templates below:
1. If you need to interact with the user:
```
Thought: xxx (description of your thought process )
Response: xxx (the content you need to inquire or reply)
```
2. If you need to call an API (only one API call per time):
```
Thought: xxx (description of your thought process )
Action: xxx (the function name to be called, do not prefix "functions.")
Action Input: xxx (the parameters for the function, must be in strictly
valid JSON format)
```
```

Listing 8: Prompt for ReAct

**Evaluation Prompts**   During the evaluation process, to ensure fairness in the results, we basically reused the prompts from FlowBench. However, for the final statistics, we only used binary results to mitigate the bias issue of the judge model (see the discussion in Sec. 5.1). Below are the prompts we used for turn-level evaluation.

```
Please serve as an impartial judge to evaluate the response quality of
the assistant. Your evaluation should be based on the following criteria:
(1) Correctness: Does the reply remain consistent with the workflow
knowledge without any contradictions?
```

```
(2) Helpfulness: Has the user's request been reasonably understood and
addressed, fulfilling the user 's needs within the provided workflow
scope?
(3) Humanness: Is the response coherent, clear, complete, and does it
include human acknowledgment?
Please compare the provided response with the reference response and
evaluate it based on the mentioned dimensions. Then, aggregate these
assessments to assign an overall score.
A perfect score is 10 points, with 9-10 points indicating high quality,
nearly identical to the reference answer; 7-8 points indicating quality
close to the reference answer; 6-7 points being of moderate quality; 4-5
points indicating a lower quality response; and 2-3 points for a response
 with significant errors.
Finally, output a binary result to determine if the predicted and
reference responses are consistent (Yes or No).

Here is the knowledge related to the workflow:
```
{{ workflow_info }}
```

Here is the previous conversation:
```
{{ session }}
```

Here is the true value response from the reference:
{{ reference_input }}

Here is the generated response from the assistant:
{{ predicted_input }}

Please reply with the scores and consistency judgment in the following
format:
```
Correctness Score: xxx
Helpfulness Score: xxx
Humanness Score: xxx
Consistency: Yes/No
```
```

Listing 9: Prompt for turn-level evaluation

# D  ADDITIONAL METHOD DETAILS

## D.1  FLOWAGENT EXECUTION FRAMEWORK

To clearly demonstrate the execution process of FLOWAGENT, we provide the pseudocode of the
FLOWAGENT execution process here.

## D.2  DATA CONSTRUCTION

Based on existing datasets, we performed data transformation and construction to evaluate agent per-
formance across the compliance and flexibility dimensions. Our data construction process consists
of three stages: *workflow collection, workflow representation*, and *dialogue construction*.

**Workflow Collection**  Our dataset comprises two existing datasets: SGD (Rastogi et al., 2019)
and STAR (Mosig et al., 2020), as well as our own constructed dataset, In-house. The SGD dataset
includes 26 task flows across 16 domains, while the STAR dataset covers 24 task flows across 13
domains. The In-house dataset, constructed manually based on real-world scenarios in business,
contains 6 workflows and 16 tools across 6 domains.

---

**Algorithm 1:** FLOWAGENT Execution Framework

---

**Input:** user $\mathcal{U}$, bot agent $\mathcal{A}^{(pdl)}$, system $\mathcal{S}$, workflow in PDL format $\mathcal{W}^{(pdl)}$, pre-decision controllers
$\quad\quad \mathcal{C}_{\text{pre}} = \{c_i^{\text{pre}}\}_{i=1}^{C_{\text{pre}}}$, post-decision controllers $\mathcal{C}_{\text{post}} = \{c_j^{\text{post}}\}_{j=1}^{C_{\text{post}}}$, maximum attempts per turn $N_{\max}$
**Output:** conversation history $\mathcal{H}$

1   Initialize conversation history: $\mathcal{H} \leftarrow \emptyset$ ;
2   **while** *True* **do**
3     $\mathcal{O}_{\mathcal{U}} \leftarrow \mathcal{U}(\mathcal{H})$ ;
4     $\mathcal{H} \leftarrow \mathcal{H} \,\|\, \mathcal{O}_{\mathcal{U}}$ ;
5     **if** $\mathcal{O}_{\mathcal{U}}.is\_end = True$ **then**
6       **break** ;
7     **for** $turn\_id \leftarrow 1$ **to** $N_{max}$ **do**
       `// Traverse all pre-decision controllers`
8       $\mathcal{R}_{\text{pre}} \leftarrow \emptyset$ ;
9       **foreach** $c_i^{pre} \in \mathcal{C}_{pre}$ **do**
10         $r_i \leftarrow c_i^{\text{pre}}.\text{process}(\mathcal{H}, \mathcal{W}^{(pdl)})$ ;
11         $\mathcal{R}_{\text{pre}} \leftarrow \mathcal{R}_{\text{pre}} \,\|\, r_i$ ;
12       $\mathcal{O}_{\mathcal{A}} \leftarrow \mathcal{A}^{(pdl)}(\mathcal{H}, \mathcal{W}^{(pdl)}, \mathcal{R}_{\text{pre}})$ ;
       `// Traverse all post-decision controllers`
13       if\_pass $\leftarrow$ True ;
14       **foreach** $c_j^{post} \in \mathcal{C}_{post}$ **do**
15         **if** $c_j^{post}.process(\mathcal{O}_{\mathcal{A}}) = False$ **then**
16          if\_pass $\leftarrow$ False ;
17       **if** $if\_pass = True$ **then**
18         **if** $\mathcal{O}_{\mathcal{A}}.type = tool\_calling$ **then**
19          $\mathcal{O}_{\mathcal{S}} \leftarrow \mathcal{S}(\mathcal{O}_{\mathcal{A}})$ ;
20          $\mathcal{H} \leftarrow \mathcal{H} \,\|\, \mathcal{O}_{\mathcal{S}}$ ;
21         **else if** $\mathcal{O}_{\mathcal{A}}.type = response\_to\_user$ **then**
22          $\mathcal{H} \leftarrow \mathcal{H} \,\|\, \mathcal{O}_{\mathcal{A}}$ ;
23          **break** ;

---

Table 4: Dataset Statistics

| | Datase | # Workflow | # Session | # Turn | # User Profile | # User Intentions | # OOW queries |
|---|---|---|---|---|---|---|---|
| | SGD | 26 | 442 | 11,594 | 390 | 1,593 | 811 |
| session-level | STAR | 24 | 408 | 10,856 | 360 | 1,265 | 679 |
| | In-house dataset | 6 | 102 | 3,246 | 90 | 322 | 212 |
| | SGD | 26 | 338 | 5,016 | - | 834 | 496 |
| turn-level | STAR | 24 | 312 | 5,387 | - | 853 | 541 |
| | In-house dataset | 6 | 150 | 1,679 | - | 353 | 203 |

**Workflow Representation**    To compare the performance of our PDL syntax with other workflow formats, we converted each workflow under investigation into four formats: natural language, code, flowchart, and PDL. Referring to Xiao et al. (2024), we first converted the workflows from the original datasets into natural language. Then, we used a LLM to respectively transform them into code,

Table 5: Comparison of Contents Included in Different Datasets

| | SGD | ABCD | STAR | FLAP | FlowBench | In-house dataset |
|---|---|---|---|---|---|---|
| Workflow Format | - | NL | flowchart | NL | NL, code, flowchart | NL, code, flowchart, PDL |
| Multiple User Intentions | ✔ | ✔ | ✔ | ✘ | ✔ | ✔ |
| Incorporate User Persona | ✘ | ✘ | ✘ | ✘ | ✔ | ✔ |
| Automate dialogue construction | ✘ | ✘ | ✘ | ✘ | ✔ | ✔ |
| OOW Query Annotation | ✘ | ✘ | ✘ | ✘ | ✘ | ✔ |

flowchart, and PDL formats. The definitions of tools (a.k.a., APIs) follows the OpenAI function calling formats.[2] The entire workflow format conversion process was completed using GPT-4-Turbo.

**Dialogue Construction**  For turn-level evaluation, we constructed diverse user intentions from tasks, using GPT-4o to directly construct reference sessions. We then parsed and annotated tool calls at the turn level. Regarding the construction of OOW scenarios, we strategically insert OOW queries into the reference session and record the OOW information.

For session-level evaluation, we selected user personas from Chan et al. (2024) that exhibits real-world diversity in response style and format. We incorporated them into workflows to construct task-related user profiles. We employed three LLMs to respectively simulate the roles of user, agent, and system with the given user profiles, workflow descriptions, and tool definitions. We collected these simulated dialogues to form the session-level evaluation dataset. As for the OOW scenarios, we have simulated users generating OOW queries with a certain probability, prompting the agent to respond to these queries and continue the conversation. The example of generated conversation is shown in App. B.1

**Dataset Statistics**  The statistics of our formatted dataset are presented in Tab. 4. In addition, Tab. 5 presents the differences between our dataset and existing workflow benchmarks.

# E  ADDITIONAL EXPERIMENTAL RESULTS

## E.1  TURN-LEVEL EVALUATION RESULTS

The table below presents the turn-level experimental results of Qwen2-72B. It's important to note that because Out-of-Workflow (OOW) turns typically involve fewer complex conditional judgments or API calls, the turn-level *Success Rate* for OOW turns can sometimes be higher than for In-Workflow (IW) turns. Additionally, since the turn-level evaluation for the OOW portion involves fewer API calls, directly calculating this metric may introduce significant variance. Therefore, we have left it blank in the table.

Table 6: Turn-level Evaluation Results of Qwen2-72B

|  | Method | In-house dataset | | | STAR | | | SGD | | |
|---|---|---|---|---|---|---|---|---|---|---|
|  |  | Pass Rate | Tool F1 | Parameter F1 | Pass Rate | Tool F1 | Parameter F1 | Pass Rate | Tool F1 | Parameter F1 |
| IW | ReAct$_{NL}$ | 65.82 | 76.71 | 65.75 | 58.66 | 65.64 | 51.02 | 60.81 | **68.02** | 58.39 |
|  | ReAct$_{code}$ | 46.83 | 55.70 | 55.44 | 49.41 | 45.81 | 42.34 | 48.95 | 55.11 | 47.52 |
|  | ReAct$_{FC}$ | 65.04 | 71.58 | 67.70 | 60.97 | 65.19 | 50.29 | 62.47 | 65.40 | 55.17 |
|  | FLOWAGENT | **68.35** | **77.14** | **68.12** | **68.94** | **67.66** | **62.19** | **64.19** | 67.65 | **60.78** |
| OOW | ReAct$_{NL}$ | 66.67 | 71.42 | - | 49.61 | 60.33 | - | 61.32 | 47.76 | - |
|  | ReAct$_{code}$ | 45.35 | 45.71 | - | 41.86 | 57.89 | - | 55.81 | 36.50 | - |
|  | ReAct$_{FC}$ | 60.07 | 74.17 | - | 51.94 | 65.00 | - | 65.89 | 68.21 | - |
|  | FLOWAGENT | **71.67** | **80.55** | - | **59.52** | **70.74** | - | **68.21** | **70.74** | - |

## E.2  VISUALIZATION OF METRIC COMPARISON

The figures below illustrate the *Task Progress* metric for GPT-4o in session-level evaluation and the *Pass Rate* metric for Qwen2-72B in turn-level evaluation. Refer to Tables 1, 2, and 6 for detailed values.

## E.3  WIKIHOW-BASED QA BACKGROUND

As a well-known online platform, WikiHow[3] provides step-by-step guides on various topics, offering users instructions for everyday tasks. The data on WikiHow naturally exhibit workflow characteristics, as the guides are organized in a sequential, stepwise format. For instance, Fig. 4(a)

---

[2]https://platform.openai.com/docs/guides/function-calling
[3]https://www.wikihow.com/

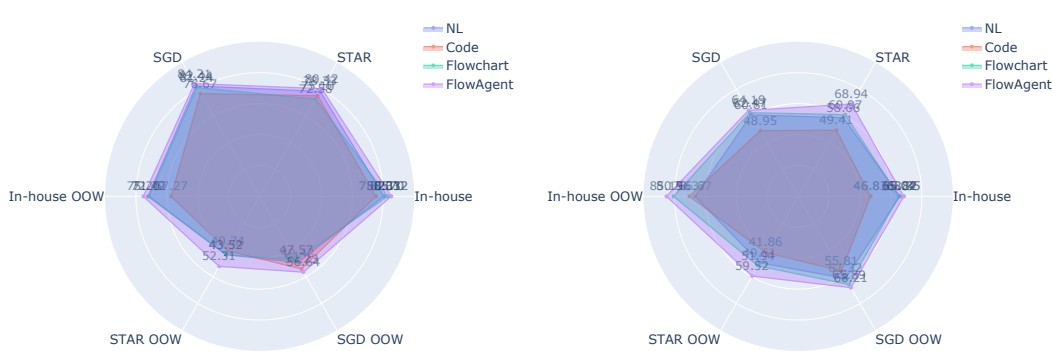

(a) *Task Progress* metric for GPT-4o in session-level evaluation.

(b) *Pass Rate* metric for Qwen2-72B in turn-level evaluation.

Figure 5: Visualization of the comparison of metrics for different models.

illustrates the structure of a WikiHow page describing four different methods for checking the publication date of a website. Each method includes a series of ordered steps, which can be considered as sub-procedures within the broader workflow. An intelligent agent could help user navigate these procedural guides more efficiently. We refer to this type of task as *workflow-based QA*, where the agent assists users in understanding and using the guides through conversation. This interactive form differs from the traditional workflow agents introduced in Sec. 3.2, as it does not involve explicit state maintenance or tool invocation.

To validate this, we first used GPT-4-Turbo to automatically reformulate data from the WikiHow website using PDL syntax, as shown in Fig. 4(b), where four distinct sub-procedures are defined as four SUBTASKs. Building on this, Fig. 4(c) demonstrates a sample dialogue between the user and FLOWAGENT, showing that the bot can follow the PDL-defined workflow and respond to user queries based on the content described in each SUBTASK. Through these preliminary examples, we observed that PDL syntax can effectively represent WikiHow-like workflows, and the FLOWAGENT framework supports this new type of workflow-based QA task. Future work will include systematic evaluation and analysis focused on refining WikiHow workflow and experimental design for this task.

