# OpenReview forum: "FlowAgent: a New Paradigm for Workflow Agent"
_ICLR.cc/2025/Conference — Submitted to ICLR 2025_

### Official Review · Reviewer_TVxY · 2024-10-31

**Soundness:** 2
**Presentation:** 2
**Contribution:** 2
**Rating:** 5
**Confidence:** 3

**Summary:**

The paper deals with an important topic: that of controlling the responses of an LLM based on a workflow. The authors introduce a new workflow specification language for controlling the LLMs’ responses, called PDL.

**Strengths:**

The proposed topic is interesting and, certainly, useful. However, several issues in the presentation make it difficult to understand the contributions of this work and the limitations of the state-of-the-art. For example, the authors talk about out-of-workflow queries, however a clear definition/example of such queries is missing.

As I state at the end of comments/questions to the authors (please see below), I am willing to increase my score if the authors address my comments.

**Weaknesses:**

*Typos*

Sentence in line 014-015 difficult to understand. Please revise.

*Presentation issues*

In the paragraph starting in line 042, the authors discuss the pros and cons of prompt-based and rule-based methods. Two comments:
- References to prompt-based and rule-based methods are missing, making difficult for the readers to understand how the aforementioned approaches work.
- A concrete example demonstrating the issues describe in the paragraph in lines 042--052 is missing (i.e., how the responses of a state-of-the-art prompt-based method and of a state-of-the-art rule-based method a state-of-the-art differ on a specific user question based for a given LLM). I recommend the authors adding such an example as that would help the readers understand concretely the limitations of previous work and appreciate the contributions of the proposed work.

*Related work*

- A more elaborated discussion on previous work is missing. The only discussion I found is in Section 2.2.
- Please add a discussion and conduct experiments against work on constrained-decoding using regular expressions, which I find it very relevant to this work:
(1) “Grammar-Constrained Decoding for Structured NLP Tasks without Finetuning” by Sabio Geng et al.
(2) “Validating Large Language Models with Relm” by Michael Kuchnik et al.

*Technical questions*

- How PDL differs from other workflow specification languages, not necessarily designed to support LLMs – some references for the authors to consider:
(1) Serge Abiteboul, Pierre Bourhis, Victor Vianu. Comparing workflow specification languages: A matter of views. ACM Transactions on Database Systems, 2012, 37 (10).
(2) W.M.P. van der Aalst and A.H.M. ter Hofstede .YAWL: yet another workflow language. Information Systems. Volume 30, Issue 4, June 2005, Pages 245-275.
- Following up from the previous question: why cannot we use/extend existing workflow specification languages to constrain the responses of LLMs?
- In line 050 the author state: “for an existing rule-based workflow, if we want it to support a new demand outside the original procedure, such as helping a user check the weather (the yellow diamond in the figure), significant modifications to the original workflow are required, which becomes impractical as more out-of-workflow demands are required.” How exactly PDL overcomes this crux? This should be demonstrated both via a (toy) example and via experiments.
- The authors say that their framework can support out-of-workflow queries, however a clear definition/example of such queries is missing, making it difficult to assess the importance of the proposed work.

*Overall assessment*

A clear positioning of the contributions against the state-of-the-art is missing, making it difficult to understand the novelty of this research. I am willing to increase my score if the authors address my comments/questions.

**Questions:**

Please see my questions/comments in the above field. In particular:

- Please give a concrete example from a domain of your interest (e.g., booking a hotel room) showing how prompt-based and rule-based methods from paragraph in 042--052. differ on their responses to a user query and stressing the limitation of the prior art and the contributions of this work. Please also give exact references-- there is no reference in this paragraph.
- Please create a table or analyse the key differences between PDL and previous established workflow languages (see for example Serge Abiteboul, Pierre Bourhis, Victor Vianu. Comparing workflow specification languages: A matter of views. ACM Transactions on Database Systems, 2012, 37 (10)), particularly focusing on aspects relevant to LLM control.
- Please provide a dedicated subsection defining out-of-workflow queries, along with a few concrete examples demonstrating how these queries differ from in-workflow queries and how PDL handles them.
- Please experimentally compare your framework against one recent constrained-decoding technique using regular expressions, e.g., “Validating Large Language Models with Relm” by Michael Kuchnik et al. The experiment should demonstrate how the authors' approach differs from or improves upon previous constrained-decoding methods in the context of workflow control for LLMs.
- Please provide a specific toy example showing how PDL handles a new out-of-workflow demand, and include an experimental comparison demonstrating the flexibility of PDL versus rule-based approaches in adapting to new demands.

---

> ### Author Response · Authors · 2024-11-26
>
> Thank you for your detailed review. We have revised the paper based on your feedback, with major updates highlighted in blue in the PDF.
>
> ### Prompt-Based and Rule-Based Methods Example (Section 3 and Appendix A.1)
>
> Thank you for the suggestion. We have revised the introduction and added references in the updated version, specifically in the second and third paragraphs. A more detailed discussion on prompt-based and rule-based workflows is now in Section 3.
>
> Using the "Book Apartment Viewing" example from the STAR dataset, we illustrate the differences between prompt-based and rule-based methods. Below is the natural language workflow (prompt-based), which describes the process for the agent:
>
> ```text
> 1. Ask the user for their name.
> 2. Request the name of the apartment the user is interested in.
> 3. Ask the user for the day they wish to view the apartment.
> 4. Ask for the start time of the viewing.
> 5. Inquire if the application fee has been paid.
> 6. Request any custom message the user might want to add.
> 7. Check the availability for the requested viewing.
> 8. If the viewing is available, inform the user and ask if they want to proceed with booking.
> 9. If the user says yes, book the viewing and inform them that the booking was successful.
> 10. If the user says no, ask again for the apartment name to modify search criteria.
> 11. If the viewing is not available, inform the user that the viewing is unavailable.
> 12. Finally, ask if there's anything else the user needs assistance with.
> ```
>
> Natural language workflows can be imprecise, especially with nested branches like lines 7-11, which may confuse LLMs.
>
> On the other hand, in a rule-based workflow, the entire process is rigidly controlled by software, and the LLM is only called as a node within this process. For example, in the Dify workflow [1], we need to configure a fixed sequence, such as using LLM_1 to complete the first step, "Ask the user for their name." Only after this step is completed can we move to the second step (LLM_2 to "Request the name of the apartment the user is interested in"). However, in reality, users may not respond according to this flow, such as saying, "Tell me first if I can book." In this case, the process gets stuck at the first step, and LLM_1 will continue to ask, "Please provide your name first".
>
> Our proposed PDL syntax is as follows:
>
> ```yaml
> Name: Book Apartment Viewing
> Desc: A process to assist users in booking a viewing for an apartment.
>
> SLOTs:
>   - name: RenterName
>   - name: ApartmentName
>   - name: Day
>   - name: StartTimeHour
>   - name: ApplicationFeePaid
>   - name: CustomMessage
>   - name: RequestType
>
> APIs:
>   - name: BookApartmentViewing
>     request: [RenterName, ApartmentName, Day, StartTimeHour, ApplicationFeePaid, CustomMessage, RequestType]
>     response: [Message]
>
> ANSWERs:
>   - name: ViewingAvailable
>     desc: Inform the user that the viewing is available and ask if they want to proceed with booking.
>     precondition: [BookApartmentViewing]
>   - name: BookingSuccess
>     desc: Inform the user that the booking was successful.
>     precondition: [BookApartmentViewing]
>   - name: ModifySearchCriteria
>     desc: Ask the user for a different apartment name to modify search criteria.
>     precondition: [BookApartmentViewing]
>   - name: ViewingUnavailable
>     desc: Inform the user that the viewing is unavailable.
>     precondition: [BookApartmentViewing]
>   - name: FurtherAssistance
>     desc: Ask if there's anything else the user needs assistance with.
>     precondition: [BookApartmentViewing]
>
> PDL: |
>   [RenterName] = ASKSLOT()
>   [ApartmentName] = ASKSLOT()
>   [Day] = ASKSLOT()
>   [StartTimeHour] = ASKSLOT()
>   [ApplicationFeePaid] = ASKSLOT()
>   [CustomMessage] = ASKSLOT()
>   [RequestType] = "CheckAvailability"
>   [Message] = API_BookApartmentViewing([RenterName, ApartmentName, Day, StartTimeHour, ApplicationFeePaid, CustomMessage, RequestType])
>   if Message == "Viewing Available":
>       ANSWER_ViewingAvailable()
>       UserWantsToProceed = ASKSLOT()
>       if UserWantsToProceed == "Yes":
>           [RequestType] = "BookViewing"
>           [Message] = API_BookApartmentViewing([RenterName, ApartmentName, Day, StartTimeHour, ApplicationFeePaid, CustomMessage, RequestType])
>           ANSWER_BookingSuccess()
>       else:
>           ANSWER_ModifySearchCriteria()
>           [ApartmentName] = ASKSLOT()
>   elif Message == "Viewing Unavailable":
>       ANSWER_ViewingUnavailable()
>   ANSWER_FurtherAssistance()
> ```
> Compared with the formal two syntaxes, PDL has the following features:
> 1. **Flexibility:** Unlike rule-based workflows, PDL allows for more flexibility, addressing issues like user deviations.
> 2. **Clarity:** Compared to natural language workflows, PDL provides a clear structure, aiding LLMs in understanding the process.
>
> [1] https://docs.dify.ai/guides/workflow

---

> > ### Author Response · Authors · 2024-11-26
> >
> > ### Differences Between PDL and Other Workflow Languages
> >
> > Thank you for highlighting relevant work on workflow languages [3,4], which include YAWL (Yet Another Workflow Language), BPEL (Business Process Execution Language), and others. To our understanding, these are part of workflow management systems (WfMS), emphasizing syntactic completeness and enabling automated process execution through a management system [1,2].
> >
> > In contrast, our paper focuses on automating workflow execution in specific domains, such as chatbots and agents, by integrating LLMs. Unlike WfMS, which uses software to precisely control workflow execution, we are concerned with creating syntax that is more suitable for LLM understanding and addressing the cost and efficiency of manually writing workflows.
> >
> > Overall, we categorize WfMS as "rule-based workflows" since they rely on pre-written software for execution, whereas PDL is considered a "prompt-based workflow" as it primarily depends on LLMs for automated understanding and execution.
> >
> > | | WfMS | PDL |
> > |---|---|---|
> > | **Goal** | Precisely describe and execute workflows | Introduce workflow to LLMs |
> > | **Application** | Business process management | LLM-based chatbot, agent |
> > | **Syntax Requirements** | Precision and completeness | Flexibility, easy for LLM understanding |
> > | **Writing Difficulty** | High, requires a learning curve | Relatively simple, user-friendly |
> > | **Installation and Usage** | Requires specific software installation (e.g., YAWL) | Directly invoke LLM |
> >
> > Currently, we haven't seen work on applying WfMS syntax to LLMs, but it's an interesting direction. It requires a deep understanding of WfMS syntax and execution software, and we plan to further explore its feasibility in future research.
> >
> > [1] https://en.wikipedia.org/wiki/Workflow_management_system
> > [2] https://en.wikipedia.org/wiki/Business_process_management
> > [3] Serge Abiteboul, Pierre Bourhis, Victor Vianu. Comparing workflow specification languages: A matter of views. ACM Transactions on Database Systems, 2012, 37 (10)
> > [4] W.M.P. van der Aalst and A.H.M. ter Hofstede. YAWL: yet another workflow language. Information Systems. Volume 30, Issue 4, June 2005, Pages 245-275. https://yawlfoundation.github.io/

---

> > > ### Author Response · Authors · 2024-11-26
> > >
> > > ### Out-of-Workflow Queries Example
> > >
> > > Using the "Book Apartment Viewing" example, we illustrate OOW queries:
> > >
> > > | OOW Category        | Description                                                                 | Example                                                   |
> > > |---------------------|-----------------------------------------------------------------------------|-----------------------------------------------------------|
> > > | Intent Switching    | Sudden changes in request details or needs, such as slot modifications or cancellations. | When asked if they want to proceed, the user says, "Change the viewing time to next Saturday." |
> > > | Process Jumping     | Deviating from the established workflow, including skipping steps or ignoring rules. | The user says, "Please book this apartment for me," before specifying a viewing date. |
> > > | Irrelevant Answering| Providing unrelated responses, such as topic shifts or counter-questions.   | When asked for their name, the user replies, "But I'm so tired."         |
> > >
> > > For IW queries, the LLM only needs to follow the workflow's instructions to take action. For OOW queries, the LLM must make more flexible decisions and respond independently to user requests. In FlowAgent, we designed two approaches to handle OOW queries:
> > >
> > > 1. For the first two types of OOW queries, we allow the LLM to autonomously determine workflow node transitions and use designed controllers to prevent serious errors (see Section 4.2 of the revised paper).
> > > 2. For the last type of OOW queries, we added "answer_oow_questions" in the node definitions to guide the LLM to select this node for responding to unrelated questions. For example, for a request like "helping a user check the weather," we added a weather API to the available APIs (not linked to other nodes). This allows the model to autonomously call the API and respond when the user needs a weather check, without adjusting other nodes in the PDL. In a rule-based workflow, supporting weather responses at any node would require adding transition edges at each node, which is cumbersome (consider the Dify implementation for better understanding).
> > >
> > > Here are two examples using the "Book Apartment Viewing" workflow to illustrate the characteristics of different methods:
> > >
> > > 1. If a user replies with "But I'm so tired," a rule-based workflow might continue with "I understand, but please provide your name to proceed," whereas the PDL-based FlowAgent would respond more kindly with "What's going on? Sometimes it's okay to take a break."
> > > 2. For natural language workflows, due to the lack of clear process descriptions in unstructured text, we found in experiments that LLMs are more likely to make incorrect transitions, such as calling the BookApartmentViewing API without collecting all required slots. The PDL-based FlowAgent helps alleviate this issue.
> > >
> > > ### Comparison with Constrained-Decoding Techniques
> > >
> > > Thank you for providing the related work. We read the paper [1], but it seems to differ from the issues we are exploring. For constrained-decoding methods, we believe their goal is to generate structured output. However, in our workflow agent scenario, we can't find a convenient way to define the output format in a structured form, such as using regular expressions (regex).
> > >
> > > A somewhat similar approach is discussed in paper [2], but it relies on a predefined, structured graph, which doesn't match our workflow setup. We will explore the possibility of using constrained-decoding for the workflow agent in future work.
> > >
> > > [1] Kuchnik, Michael, Virginia Smith, and George Amvrosiadis. "Validating large language models with relm." arXiv preprint arXiv:2211.15458 (2022).
> > > [2] Roy, Shamik, et al. "Flap: Flow adhering planning with constrained decoding in llms." arXiv preprint arXiv:2403.05766 (2024).
> > >
> > > ### Additional Adjustments
> > >
> > > 1. We added a Related Work section to help readers understand the context of our research.

---

### Official Review · Reviewer_MtpL · 2024-11-02

**Soundness:** 2
**Presentation:** 3
**Contribution:** 3
**Rating:** 5
**Confidence:** 4

**Summary:**

The authors propose FlowAgent, an agent built using LLMs and incorporates workflow rules to balance between flexibility and compliance. The agent leverages both pre-decision and post-decision controllers to adjust the agent's behavior. Experimental results on existing and custom datasets show FlowAgent's superiority compared to other approaches from the literature.

**Strengths:**

The paper was easy to read and addresses an interesting problem in the literature that is very relevant to industry applications. The solution also seems simple to implement making it easy to adopt in use cases. The experimental section shows the benefits of the authors' proposed method.

**Weaknesses:**

The paper contains a few gaps in the both the presentation of the work and the experimental section.

First, it is unclear how the controllers were implemented, i.e., there isn't sufficient information in the write up for one to reimplement or even have an idea of what approach was adopted. It seems that the controllers rely on deterministic syntax and logic checks but that is pure speculation on my part as the paper only has a few lines describing both pre and post decision controllers, namely focusing on their purpose as opposed to their implementation. It would be helpful if authors could either include a pseudocode of their controllers or a diagram describing their operations.

Second, the experimental section does not discuss the computational cost and additional overhead of FlowAgent compared to other approaches. Since FlowAgent operates in a conversational setting, a level of responsiveness is expected but we have no way of knowing from the paper how much overhead the added controllers are causing from both responsiveness and computational cost perspectives. It would be helpful if the authors included runtime comparisons or latency measurements between FlowAgent and the baseline approaches in their experimental results. Additionally, an analysis of how the controllers impact the overall response time in a conversational setting and the accuracy of the end to end system would be very helpful

While figure 2 serves as the main architecture diagram describing the approach, it is very abstract. The paper could benefit from another figure showing how the controllers were working since those seem to be a key contributor. Again, I assume they rely on some sort of graph algorithm based on the DAG in figure 3 but more information is needed. Including a flowchart showing the decision-making process of the pre-decision and post-decision controllers, or a diagram illustrating how they interact with the DAG structure. This would help clarify the relationship between the controllers and the workflow representation.

**Questions:**

NA

---

> ### Author Response · Authors · 2024-11-26
>
> Thank you for your detailed review. We have revised the paper based on your feedback, with the main updates highlighted in blue in the PDF. Below, we address your concerns point by point:
>
> ### Implementation Details of Controllers
>
> We apologize for the initial lack of detail. We have added more information in Section 4.2 of the revised paper and included Algorithm 1 to demonstrate the operation of the controllers. Here’s a brief response to your questions:
>
> 1. The controllers are indeed based on deterministic logic, implemented in Python.
> 2. Each pre-decision controller checks the current state and organizes the results into the LLM’s prompt. Each post-decision controller checks the LLM-generated decisions to prevent undesired outcomes. We have provided pseudocode to illustrate this process in Algorithm 1.
> 3. For clarity, we have simplified the relevant code, shown in Python below.
>
> ```python
> for i in range(agent_max_retry):
>     # Pre-decision controllers: traverse and use their messages to generate the prompt
>     pre_messages = []
>     for c in pre_controllers:
>         pre_messages.append(c.process())
>     prompt = gen_prompt(template, pre_messages, ...)
>     # Agent generates action
>     agent_output = agent.process(prompt, ...)
>     # Post-decision controllers: traverse and check the agent's action
>     if_pass = True
>     for c in post_controllers:
>         post_result = c.process(bot_output)
>         if not post_result:
>             if_pass = False
>             break
>     if if_pass: # Break the loop if all post-decision controllers pass
>         break
> ```
>
> ### Performance Metrics
>
> Thank you for highlighting the importance of performance metrics. Here’s how we address your concerns:
>
> 1. **LLM Performance**: We focus on TTFT (Time To First Token) and Total Time (from request to final token). These metrics are influenced by the length of prompts and responses. The table below summarizes the average token counts and timing metrics for FlowAgent and three other syntaxes on our in-house dataset. Despite FlowAgent having longer prompts, the impact on TTFT is minimal. Since all approaches use the ReAct format for output, their output lengths are similar, resulting in comparable Total Time.
> 2. **Controllers**: As they are simple Python scripts, their computational cost is negligible compared to the LLM. In our scenarios, the number of graph nodes is typically less than 100, and the time to run checking algorithms is under 0.2ms, making it insignificant relative to LLM latency.
>
> | Format               | NL    | Code  | FC    | FlowAgent |
> |----------------------|-------|-------|-------|-----------|
> | Avg #tokens of prompt| 791.2 | 1236.7| 843.2 | 1556.5    |
> | Avg #tokens of response | 61.1 | 70.8  | 71.2  | 62.5      |
> | TTFT                 | 220ms | 251ms | 235ms | 262ms     |
> | Total Time           | 2.52s | 2.77s | 2.85s | 2.60s     |

---

> > ### Author Response · Authors · 2024-11-26
> >
> > ### Figure 2 and Graph Algorithms
> >
> > Thank you for your suggestion. We have made the following changes:
> >
> > 1. To clarify FlowAgent’s operation, we added Algorithm 1 alongside Figure 2 for a more rigorous explanation of our process.
> > 2. For the graph algorithms used in the controllers, we have organized the relevant code below. Here’s a brief complexity analysis: Building the graph in `NodeDependencyController` has a time complexity of \(O(V)\), where \(V\) is the number of nodes. The `get_invalid_node_names` method called in `pre_control` traverses the graph, checking invalid nodes with a complexity of \(O(E+V)\), where \(E\) is the number of edges. The `post_check` method verifies all prerequisite nodes for the current action with a complexity of \(O(V)\). As noted,  the number of nodes in practical workflows are not very large, so the controller's delay is negligible compared to LLM latency.
> >
> > ```python
> > class NodeDependencyController(BaseController):
> >     def _build_graph(self, pdl:PDL):
> >         g = PDLGraph()
> >         for node in pdl.nodes:
> >             node = PDLNode(name=node["name"], preconditions=node.get("precondition", None))
> >             g.add_node(node)
> >         self.graph = g
> >
> >     def pre_control(self) -> str:
> >         next_turn_invalid_apis = self.graph.get_invalid_node_names()
> >         if len(next_turn_invalid_apis) > 0:
> >             msg = f"APIs {next_turn_invalid_apis} dependency not satisfied: The precondition API has not been called."
> >         else:
> >             msg = ""
> >         return msg
> >
> >     def post_check(self, bot_output:BotOutput) -> bool:
> >         next_node = bot_output.action
> >         node = self.graph.name2node[next_node]
> >         if node.precondition:
> >             for p in node.precondition:
> >                 if not self.graph.name2node[p].is_activated:
> >                     add_system_message(f"Precondition check failed! {p} not activated for {next_node}!")
> >                     return False
> >         node.is_activated = True
> >         add_system_message(f"Check success! {next_node} activated!")
> >         return True
> >
> > class PDLGraph:
> >     def get_invalid_node_names(self) -> List[str]:
> >         invalid_nodes = []
> >         for node in self.name2node.values():
> >             for precondition in node.precondition:
> >                 if not self.name2node[precondition].is_activated:
> >                     invalid_nodes.append(node)
> >                     break
> >         return [node.name for node in invalid_nodes]
> > ```

---

> > ### Comment · Reviewer_MtpL · 2024-11-27
> >
> > Thank you for addressing my concerns. Your responses are very helpful.

---

### Official Review · Reviewer_Vqiv · 2024-11-02

**Soundness:** 2
**Presentation:** 2
**Contribution:** 2
**Rating:** 3
**Confidence:** 5

**Summary:**

The paper creates a framework (FlowAgent) for task-oriented agents that can offer flexibility and compliance with LLMs. They propose a new language, PDL, that creates a Directed Acyclic Graph with out-of-workflow (OOW) slots, making the agent flexible and using pre- and post-controllers for compliance. They perform an extensive evaluation on augmented datasets for showing how PDL handles flexibility and compliance.

**Strengths:**

- The work tries to add flexibility and compliance to the conversational agents, which is a real-world challenge.
- They experimented with three augmented datasets and showed that FlowAgent with PDL designed by humans works the best compared to NL, Code, and flowcharts created by GPT-o.
- The writing is clear and easy to understand.

**Weaknesses:**

- The paper has several missing details, especially regarding the experimental setup. I don't think there is enough evidence to suggest that PDL works for the WikiHow case study.
- The authors simulate users. However, as several studies have suggested, simulated users still do not capture real-world cases. The authors should have done a real-user study.
- With PDL, as the number of tools scale, it might be difficult for the developer to define all the pre_conditions. Especially since you need to define multiple flows and the logic is non-trivial to write to cover all the cases. See question 9.

**Questions:**

1. For pass rate in turn-level evaluation, do you use an LLM to check whether the output is correct by turning it into a binary classification problem?
2. Why do you use three types of OOW categorization? Changing their previous answers could be another scenario of OOW.
3. I believe that the Star dataset also contains OOW scenarios; why did you add more such dialogues, and how do you generate such queries?
4. Can you provide some stats or descriptions for your in-house dataset?
5. It is unclear how you converted existing flows into natural language, code, and flowcharts. Who made these conversions? Xiao et al. 2024 uses GPT to convert the text into NL, code, and flowcharts, whereas you write your own PDL. I don't think the comparison is fair here.
6. How do you construct reference sessions from tasks for turn-level evaluation using GPT-4o? Please provide more information.
7. Can you please provide an example conversation with the simulated users? There are works that demonstrate that user simulation is non-trivial in a conversational setting. [1]
8. Do you have any experiments with users or simulated users to suggest that PDL can handle WIKIHOW like examples?
9. In the WIKIHOW example, what would happen if the user directly asks how to find a website's publication date using code? Since the write PDL says `if publication_date is None:` then use Google search or other tools. In this, the publication date will not be None, right? This makes me believe that you must define several "flows" for the PDL to work in all scenarios.
10. Table 1 typo, for # Turn in Star turn-level row.
11. Who add the OOW nodes to the DAG?
12. What metric did you use to decide the flexibility and compliance score in Figure 1 (c)? What is the scale for the plot?

[1] Zhou, X., Su, Z., Eisape, T., Kim, H., & Sap, M. (2024). Is this the real life? is this just fantasy? the misleading success of simulating social interactions with llms. arXiv preprint arXiv:2403.05020.

---

> ### Author Response · Authors · 2024-11-26
>
> Thank you for your thorough review and insightful comments. We have revised the paper based on your feedback, with the main updates highlighted in blue in the PDF. Below, we address your major points and provide detailed responses to your questions.
>
> ## Clarifications
>
> ### C1: Using LLMs for User Simulation (Section 5.1)
> This relates to Weakness 2 and Question 7. (see Appendix B.1 for generated conversation and Appendix C.2 for our pormpts)
>
> We appreciate your observation regarding the use of LLMs for user simulation. We recognized that using LLMs to simulate users might not fully capture real-world interactions, as they tend to behave as overly cooperative users. To address this, we implemented the following strategies:
>
> 1. **Diverse User Profiles:** We generated diverse user profiles in stages. Initial personas were based on [1], with added details such as gender, job, interactive patterns (e.g., impatient or distracted), and user intentions.
> 2. **Response Diversity:** During interactions, we randomly introduced additional instructions (OOW requests, see #C2) based on these profiles.
>
> See the "Conversation Example on STAR" below, where in line 16, the LLM-simulated user employs an `intent_switching/detail-switching` strategy by modifying the appointment time.
>
> ### C2: OOW Setup and Data Generation
> This relates to Questions 2, 3, and 11.
>
> The OOW categories (Section 5.2) were derived from analyzing historical real-world user interactions with a traditional TOD system. We identified and summarized the following three categories:
>
> 1. **Intent Switching:** Sudden changes in request details or needs, such as slot modifications or cancellations.
> 2. **Process Jumping:** Deviating from the established workflow, including skipping steps or ignoring rules.
> 3. **Irrelevant Answering:** Providing unrelated responses, such as topic shifts or counter-questions.
>
> OOW data generation involves two modes:
>
> - **Turn-level:** We use a single LLM to generate a complete dialogue ("script" mode in [0]), randomly truncate it, introduce OOW queries, and continue the conversation to create reference sessions.
> - **Session-level:** LLMs simulate users ("agent" mode in [0]), as discussed in #C1.
>
> ### C3: WikiHow Setup and Experiments (Section 6.5 and Appendix E.3)
> This relates to Weakness 3 and Question 8.
>
> Our extension to WikiHow stems from a business need to handle user-uploaded product manuals for QA, while respecting privacy. We tested on randomly selected 20 pages from four categories: Cars & Other Vehicles, Computers and Electronics, Finance and Business, and Food and Entertaining [3]. We converted these into PDL format and conducted preliminary experiments. Manual testing showed that models could respond accurately to user queries under PDL syntax (see "Conversation Example on WikiHow"). However, we acknowledge the lack of established benchmarks for "workflow-based QA" and are working on building a standard benmark and an interaction environment.
>
> Regarding Weakness 3, we note:
>
> 1. No pre_conditions were set for WikiHow as LLMs do not invoke tools in this setting, avoiding the issue you mentioned.
> 2. Defining pre_conditions for complex scenarios is challenging, and we plan to explore model performance in sparse DAG settings to test PDL robustness. (See the "Intuition and Effectiveness of PDL" section of our response to #mPiU)

---

> > ### Author Response · Authors · 2024-11-26
> >
> > ## Responses to Your Questions
> >
> > 1. **Turn-level Evaluation:** Yes, we use LLMs to compare model responses with ground truth, outputting a binary result to assess decision accuracy. (Prompts in Appendix C.2)
> > 2. **OOW Categorization:** See #C2. "Changing previous answers" falls under intent switching.
> > 3. **STAR Dataset:** See #C2, the main differences between us and STAR include: 1) We have made a more detailed classification of users' out-of-flow responses; 2) We have added an OOW label in the generated dialogue turns (see line 16 in the "Conversation Example on STAR" below). This setup helps us conduct more detailed performance evaluations.
> > 4. **In-house Data:** Our data comes from real business needs, including web chatbots and phone services, with six representative scenarios. Data is anonymized and includes 102 dialogue sessions with 3,246 turns (see Table 1).
> > 5. **Data Format Conversion:** Similar to Xiao et al. 2024, we used GPT to convert workflows from text to other formats, including PDL.
> > 6. **Reference Sessions Construction:** See #C1 and #C2. For IW sessions, we use workflow and user profiles to generate dialogues; for OOW sessions, we introduce OOW queries and continue generation.
> > 7. **Simulated User Example:** See #C1. The "Conversation Example on STAR" provides an example.
> > 8. **WikiHow-like Experiments:** See #C3 for discussion.
> > 9.  **WikiHow Conversations:** See #C3. The "Conversation Example on WikiHow" demonstrates handling your scenario. FlowAgent's descriptions are guides, allowing LLMs to autonomously choose nodes like "SearchSourceCode" for questions such as "find a website's publication date using code" (a "process jumping" scenario in #C2).
> > 10. **Table 1 Typo:** Corrected, thank you.
> > 11. **OOW Nodes in DAG:** See #C2. For OOW categories 1 and 2, LLMs decide autonomously during dialogue. For category 3, we added an `answer_oow_questions` node in each PDL, unconnected to other nodes.
> > 12. **Figure 1(c) Metrics:** The figure is a schematic based on empirical observations from STAR dataset experiments, with comparisons of different workflow formats informed by [2]. Compliance and flexibility refer to session-level pass rates under IW and OOW settings.
> >
> > [0] Zhou, X., Su, Z., Eisape, T., Kim, H., & Sap, M. (2024). Is this the real life? Is this just fantasy? The misleading success of simulating social interactions with LLMs. arXiv preprint arXiv:2403.05020.
> > [1] Xin Chan, Xiaoyang Wang, Dian Yu, Haitao Mi, and Dong Yu. Scaling synthetic data creation with 1,000,000,000 personas. arXiv preprint arXiv:2406.20094, 2024.
> > [2] Rui Xiao, Wen-Cheng Ma, Ke Wang, Yuchuan Wu, Junbo Zhao, Haobo Wang, Fei Huang, and Yongbin Li. Flowbench: Revisiting and benchmarking workflow-guided planning for LLM-based agents. ArXiv, abs/2406.14884, 2024.
> > [3] https://www.wikihow.com/Special:CategoryListing#AllCategories

---

> > > ### Author Response · Authors · 2024-11-26
> > >
> > > ## Examples
> > > ### Conversation Example on STAR
> > > ```text
> > > ...
> > > BOT: Great, Alex. What’s the name of the apartment you’re interested in?
> > > USER: Maple Apartments.
> > > BOT: When would you like to view the Maple Apartments?
> > > USER: I’d like to do it on Friday.
> > > BOT: What time on Friday would you prefer for the viewing?
> > > USER: How about 3 PM?
> > > BOT: Have you paid the application fee?
> > > USER: Yes, I have.
> > > BOT: Would you like to add any custom message for the viewing?
> > > USER: No, that's fine.
> > > BOT: <Call API> book_apartment_viewing({'RenterName': 'Alex', 'Name': 'Maple Apartments', 'Day': 'Friday', 'StartTimeHour': '15', 'ApplicationFeePaid': 'Yes', 'Message': '', 'RequestType': 'CheckAvailability'})
> > > SYSTEM: {'Status': 'Available'}
> > > BOT: The viewing is available. Would you like to proceed with booking?
> > > USER: Actually, I might need to change the time. Can we do 4 PM instead?
> > >     (OOW type) intent_switching/detail-switching
> > > BOT: <Call API> book_apartment_viewing({'RenterName': 'Alex', 'Name': 'Maple Apartments', 'Day': 'Friday', 'StartTimeHour': '16', 'ApplicationFeePaid': 'Yes', 'Message': '', 'RequestType': 'CheckAvailability'})
> > > SYSTEM: {'Status': 'Available'}
> > > BOT: The new time at 4 PM is also available. Shall I book it for you?
> > > USER: Yes, please book it.
> > > ```
> > >
> > > ### WikiHow-based PDL Example
> > > ```yaml
> > > Name: Find the Publication Date of a Website
> > > Desc: A guide to finding the publication date of a webpage
> > >
> > > SUBTASKs:
> > >   - name: CheckPageAndURL
> > >     steps:
> > >       - Look underneath the headline of an article or blog post for the publication date.
> > >       - Check the bottom of the web page for a copyright date.
> > >       - See if the date is part of the URL.
> > >       - Look at the timestamps on any comments to get an estimate of the publication date.
> > >   - name: UseGoogleSearch
> > >     ...
> > >   - name: UseWaybackMachine
> > >     ...
> > >   - name: SearchSourceCode
> > >     steps:
> > >       - Right-click the web page and select `View Page Source`.
> > >       - Press `Ctrl+F` (Windows) or `⌘ Command+F` (Mac).
> > >       - Search for the term “date" or "published".
> > >       - Look for the date listed in year-month-day order.
> > >
> > > PDL: |
> > >   [publication_date] = SUBTASK_CheckPageAndURL()
> > >   if publication_date is None:
> > >     [publication_date] = SUBTASK_UseGoogleSearch()
> > >   if publication_date is None:
> > >     [publication_date] = SUBTASK_UseWaybackMachine()
> > >   if publication_date is None:
> > >     [publication_date] = SUBTASK_SearchSourceCode()
> > > ```
> > >
> > > ### Conversation Example on WikiHow
> > > ```text
> > > U: How to find a website's publication date using code?
> > > B: To find a website's publication date, you can inspect the page source and search for terms like "date" or "published" to locate the date in the metadata.
> > > U: How to inspect the page source?
> > > B: To inspect the page source, you can right-click the webpage and select "View Page Source" in your web browser.
> > > U: I got "<meta property="article:published_time" content="2024-11-20T17:45:37.542Z"/><meta property="article:modified_time" content="2024-11-20T17:45:37.542Z"/><meta property="article:author" content="Ronan Farrow"/>"
> > > B: Great! The `<meta>` tags you found contain the publication and modification dates for the article. The `article:published_time` shows when the article was first published: 2024-11-20.
> > > ...
> > > ```

---

> ### Comment · Reviewer_Vqiv · 2024-12-01
>
> Thank you for your detailed response and updated manuscript! It resolves some of my concerns about experimental setup. I still have concerns about this work.
> - I believe that since the baselines have been automatically converted using GPT and the PDL is manually crafted, the evaluations are unfair.
> - For future versions of this manuscript, the authors should add a scale to Figure 1 (c) plot. It is good for a product presentation, but the plot is not "scientific" without scale and metrics.
> - It is still unclear to me how the LLM adds OOW nodes to the DAG.
> - The authors perform "manual interaction" with the wikiHow agent, but they do not perform any experiment that suggests the broad applicability of FlowAgent to real-world workflow-based QA tasks. I highly suggest the authors run human studies for session-level evaluation (in addition to LLM-simulated users) and evaluating applicability of flowagent to real-world applications.
> - I also have reproducibility concerns over the work. The authors provide no code or data which can be used to reproduce the results.

---

> > ### Author Response · Authors · 2024-12-03
> >
> > Thank you for your feedback. Here are our responses to your concerns:
> >
> > 1. **Regarding the fairness of the experiments**: In the SGD and STAR datasets, the PDL format, like other formats (code, FlowChart), is automatically converted from text using GPT. For the in-house data, we manually created both natural language and PDL format data, then used GPT to convert them into code and FlowChart formats, followed by manual review and correction.
> > 2. **Regarding Figure 1(c)**: We will incorporate precise experimental metrics in future versions of the manuscript to present the results more scientifically.
> > 3. **Regarding OOW nodes**: For the three types of OOW queries mentioned in Clarification 2, only the third type corresponds to a node in the PDL syntax. The other two types involve transitions between predefined nodes and do not add new nodes. Therefore, in the PDL, only the `answer_oow_questions` node is added, as shown below. This is not generated by the LLM but is automatically added during post-processing. In the node dependency DAG, since this node has no precondition constraints, it does not have dependency edges with other nodes.
> >     ```yaml
> >     ANSWERs:
> >     ...
> >     - name: answer_OOW_questions
> >       desc: Respond to user questions that are out of the workflow
> >     ```
> > 4. **Regarding manual testing**: First, concerning the "real-world applications" of workflow-based QA, we have already applied it to manual-related QA tasks in practice. Second, thank you for your suggestion. We will promptly develop a standard test set based on WikiHow and conduct human experiments to obtain quantifiable metrics.
> > 5. **Regarding reproducibility concerns**: We have open-sourced the code on an Anonymous GitHub repository, and the data will be made publicly available after review.

---

### Official Review · Reviewer_mPiU · 2024-11-03

**Soundness:** 2
**Presentation:** 3
**Contribution:** 3
**Rating:** 5
**Confidence:** 4

**Summary:**

This paper introduces FlowAgent, a framework for integrating workflows into LLMs that balances procedural compliance with flexibility, addressing the limitations of existing prompt-based and rule-based methods. To achieve this, the authors propose a Procedure Description Language (PDL) that combines the natural language flexibility and programming-like precision needed for diverse workflows.

**Strengths:**

1. This paper defines a novel PDL to address OOW requests that agents may encounter during workflow execution. A comprehensive and unified framework like this facilitates further research in this field.
2. The authors also introduce a new evaluation framework specifically designed to assess workflow agents' performance in OOW scenarios.

**Weaknesses:**

1. The introduction of PDL lacks thorough analysis. After observing the impacts of OOW queries, it is unclear what specific considerations led to the development of PDL. Additionally, the completeness of PDL requires further examination to demonstrate its capability to handle more complex, real-world workflows effectively.

2. The evaluation of experiments is incomplete; the authors only assess GPT-4 and Qwen2-72B, with a brief note that “weaker models could not handle more complex workflow tasks.” However, there is no detailed analysis on what specific issues smaller models faced. Further exploration is needed to show how smaller models perform on simpler workflow tasks to provide a clearer picture of model scalability across task complexity.

3. Several key details are missing, such as the hyperparameters used during LLM inference, the prompts employed during data collection, dataset construction details, and relevant examples from the datasets. Including these would improve the reproducibility and clarity of the experiments.

**Questions:**

Minor Issues:
1. The text in Figure 4 is difficult to read, and the radar chart lacks specific performance values, making it challenging to interpret the results accurately.

---

> ### Author Response · Authors · 2024-11-26
>
> Thank you for your detailed review. We have revised the paper based on your feedback, with the main changes highlighted in blue in the PDF. Below, we address your concerns point by point:
>
> ### Intuition and Effectiveness of PDL (revised Section 4.1)
>
> **Design Principles of PDL:**
> We apologize for not clearly explaining the design principles of PDL in the initial submission. We have revised the relevant sections and provide a brief explanation here, focusing on compliance and flexibility:
>
> 1. **Flexibility:** To enhance the flexibility of workflow agents, we adopted a prompt-based approach for applying PDL (see Section 3.2). Unlike other workflow representations (natural language, code, flowchart), which cannot perceive the state of the workflow, PDL uses explicit, structured node definitions. This allows the workflow agent to detect its current state during execution.
> 2. **Compliance:** In PDL's node design, we introduced dependency nodes to define state relationships during workflow execution. Based on this, we designed a series of controllers to regulate agent behavior, enhancing compliance during the workflow process (see Sections 4.1 and 4.2).
>
> **PDL Performance in Complex Scenarios:**
>
> 1. **Empirical Analysis:** In preliminary experiments, we conducted an empirical analysis comparing PDL with other workflow representations. We found that in more complex workflows (20+ APIs, 10+ branching nodes), FlowAgent using PDL syntax, along with control modules, executed tasks more accurately than those using natural language, code, or flowchart. Execution success rates increased from below 50% to over 65%.
>
> 2. **Performance Metrics:** We divided the SGD dataset workflows into three subsets based on the number of lines of workflow code: `(0,15]`, `(15,30]`, and `(30,60]`, representing different complexity levels. The table below shows model performance using PDL and code syntax across these subsets. PDL shows more stable performance on complex workflows compared to code.
>
> | Syntax | Metric        | (0,15] | (15,30] | (30,50] |
> |--------|---------------|--------|---------|---------|
> | PDL    | Success Rate  | 34.62  | 32.15   | 33.52   |
> | PDL    | Task Progress | 84.50  | 80.12   | 82.15   |
> | PDL    | Tool          | 90.25  | 85.21   | 88.70   |
> | Code   | Success Rate  | 32.10  | 28.70   | 25.25   |
> | Code   | Task Progress | 82.20  | 76.56   | 70.18   |
> | Code   | Tool          | 88.72  | 85.61   | 78.20   |
>
> ### Performance of Different Model Sizes
>
> Thank you for the suggestion. We have added results for different sizes of the Qwen2 series models. Using the same method to divide the SGD dataset, we evaluated the Task Progress metric. The results are shown below:
>
> | Model     | (0,15] | (15,30] | (30,50] |
> |-----------|--------|---------|---------|
> | Qwen2-0.5B| 0.00   | 0.00    | 0.00    |
> | Qwen2-1.5B| 2.20   | 3.31    | 0.00    |
> | Qwen2-7B  | 42.17  | 37.40   | 35.22   |
> | Qwen2-72B | 72.23  | 69.70   | 70.42   |
>
> Observations:
> 1. Smaller models (0.5B, 1.5B) struggle with this task.
> 2. The 7B model partially understands the task but shows a significant performance gap compared to the 72B model, especially on more complex workflows.
> 3. Further analysis reveals issues with the model's ability to adhere to the specified format and accurately invoke tools when needed. We plan to optimize smaller models' performance regarding these issues in future experiments.
>
> ### Experimental Details (Appendix A, B and C)
>
> We have added detailed experimental information in the appendix of the updated paper:
>
> 1. **Hyperparameters:** For the GPT series, we used models `gpt-4o-2024-05-13`, `gpt-4o-mini-2024-07-18`, and `gpt-4-turbo-2024-04-09`, with a temperature set to 0.2 for stable outputs. For the Qwen2 series, we used the vllm framework, also with a temperature of 0.2.
> 2. **Prompts:** Key prompts used are included in the appendix, covering user simulation prompts during dataset construction, prompts for FlowAgent and baselines during inference, and adjusted prompts for evaluation based on FlowBench.
> 3. **Dataset Examples:** We added more dataset examples in the appendix, including a complete PDL workflow, a user profile example, and dialogue examples from the STAR and WikiHow datasets.
>
> We hope these additions enhance the paper's clarity and reproducibility.
>
> ### Figure 4
>
> Based on your suggestion, we have revised Figure 4. (now is Figure 5 in Appendix E.3)

---

> > ### Comment · Reviewer_mPiU · 2024-11-28
> >
> > Thanks for your reply, I'll keep my score unchanged

---

### Meta-Review · Area_Chair_Uhqs · 2024-12-21

**Metareview:**

(a) Scientific Claims and Findings
The authors propose PDL, a novel syntax for workflow expression, which integrates flexibility and precision.
A framework is introduced to address OOW queries, using pre- and post-decision controllers for regulatory compliance.
Experiments demonstrate superior performance of FlowAgent compared to natural language, code, and flowchart-based workflows on benchmark datasets.
(b) Strengths
- The introduction of PDL and its compatibility with DAG structures provides a structured yet flexible workflow representation.
- Empirical results highlight FlowAgent's superior handling of OOW queries compared to baselines.
- The paper is well-written, with clear descriptions of the problem, solution, and evaluation.
(c) Weaknesses and Missing Elements
- Reproducibility: Key details, such as prompts and experimental settings, were initially missing, though later addressed in the rebuttal.
- Baselines: Evaluation fairness was questioned since PDL data was manually curated while baselines were automated.
- Scalability Concerns: The potential difficulty in scaling PDL definitions for workflows with numerous tools or complex preconditions was highlighted.
- Real-World Validation: Absence of real-user studies limits the applicability evidence for broader, real-world settings.
- Technical Exposition: Insufficient details on pre- and post-decision controllers' implementation and computational costs.
(d) Decision: rejection
While the paper addresses an important topic and introduces a novel approach with empirical validation, several concerns about reproducibility, scalability, and applicability in real-world scenarios remain unresolved. The limitations in baselines and absence of rigorous human studies further weaken the submission. I wish the authors can incorporate the feedback and look forward to a stronger submission.

**Additional Comments On Reviewer Discussion:**

During the rebuttal period, the authors made several clarifications and revisions:

- Reproducibility:
The authors added pseudocode, detailed experimental setups, and datasets in the appendix to address reproducibility concerns. However, some reviewers noted that these updates came late and felt insufficient in ensuring the work could be independently validated.

- Experimental Fairness:
The authors clarified that PDL workflows were curated similarly to other baselines, albeit manually reviewed. This raised concerns about evaluation bias, particularly since FlowAgent's performance could be overly optimized for this setup.

- Real-World Validation:
The absence of real-user studies remained a major critique despite the authors' efforts to enhance simulated user diversity and clarify WikiHow experiments.

- Scalability:
Concerns about defining preconditions for complex workflows were partially addressed through examples but left unresolved for larger, real-world workflows.

Final Decision Considerations:
- Reviewer mPiU kept the score unchanged due to incomplete handling of scalability and experimental fairness issues.
- Reviewer Vqiv acknowledged improvements but upheld reproducibility and fairness concerns.
- Reviewer MtpL appreciated the updates on controllers but noted remaining gaps in computational cost analysis.
- Reviewer TVxY appreciated added clarity on related work but maintained the need for further comparative experiments.

---

### Decision · Program_Chairs · 2025-01-22

Reject